# A Bio-Inspired Sound Localization Spiking Neural Network with Unsupervised Local Plasticity and Proximity Learning

## Abstract

In this paper, we propose an unsupervised learning principle that leverages the neuro-inspired local plasticity and biophysiological characteristics of the brain for the learning of spiking neural networks (SNNs) without labels. The learning principle synergistically combines morphological features and biochemical phenomena in the brain cortex, guiding networks to self-organize their connectivity without global error backpropagation. The learning principle is based on two local plasticity rules. One is latency-mediated spike timing-dependent plasticity, formulated by combining the original STDP with axonal latency. The other is proximity learning, mediated by the volume transmission of neurotransmitters among neurons. We successfully applied these plasticity rules to a spiking model of the avian auditory cortex and observed the self-organization of the network, which results in the accurate localization of sound sources. After being trained using interaural time difference (ITD)-encoded spike trains, the network converged to synaptic connectivity resembling the famous Jeffress model. The performance evaluation results presented demonstrate that the proposed learning principle enables the SNN to localize sound sources with accuracy and resolution higher than those achieved by supervised learning rules.

## 1 Introduction

Sound source localization (SSL) is a highly demanded task. Its diverse applications include automatic speech recognition (ASR) (Dávila-Chacón et al. (2018)), hearing aids (Farmani et al. (2017); Fejgin & Doclo (2023)), human-robot interaction (HRI) (Rascon & Meza (2017); Li et al. (2016)), and event detection (Li et al. (2012); Bandi et al. (2012)). The intensive computational requirements associated with these applications necessitate an efficient sound source localization. Given this situation, research on approaches using bio-inspired methods, instead of the traditional SSLs based on digital signal processing (DSP), is actively pursued (Dávila-Chacón et al. (2018); Pan et al. (2021); Glackin et al. (2010); Joo et al. (2024); Rahaman & Kim (2020)). A biological SSL, illustrated in Fig. 4 in the Appendix, detects the location of the sound source using the sound input to both ears, whose mechanism based on the Jeffress model (Fig. 4b) is well understood. The Jeffress model (Jeffress (1948)) mimicking the biological SSL in the auditory cortex can be used as the basis for an artificial SSL network to mitigate the computational burden of the traditional approaches.

A biological brain, which performs functions like SSL, can be considered a spiking neural network (SNN), whose functional structure is formed by allowing its connectivity weights to be modified by repeated spiking events. Spike timing-dependent plasticity (STDP) is an unsupervised learning mechanism in the biological brain that induces changes in long-term synaptic weights, thereby facilitating the learning of neural networks. Due to its local nature, the learning rule is highly efficient, requiring only the pre- and postsynaptic spike timings (Bi & Poo (1998); Zhang et al. (1998)), which substantially reduce the computational burden and energy consumption. Using the STDP, the network can properly configure its connectivity to accurately perform cognitive tasks. A series of papers describe research results supporting the effects of the unsupervised STDP on establishing meaningful network structures (Gilson et al. (2009)). By potentiating or depressing the synaptic weights, STDP can influence the formation of network structures.

The original STDP, an unsupervised local learning inspired by biological findings (Bi & Poo (1998); Zhang et al. (1998); Hebb (2005)), offers several advantages, such as locality, energy efficiency, and computational efficiency, as well as compatibility with neuromorphic hardware. However, unsupervised STDP lacks the global error correction mechanism, which often results in unstable learning or low accuracy (Zenke & Gerstner (2017)). Recent research indicates that using unsupervised STDP-based models generally fails to achieve the high accuracy of supervised learning methods (Liu et al. (2021)). Due to this limitation, studies have increasingly focused on supervised STDP that can provide guidance toward optimal states and introduce the global error correction (Liu et al. (2021); Wu et al. (2008); Muleta & Kong (2025)). Many of these approaches incorporate the backpropagation (BP)-based methods with teaching signals to enhance the performance (Wu et al. (2008); Araki & Hattori (2023); Tavanaei & Maida (2019)). While the supervised STDP achieves higher accuracy, this type of learning requires labeled datasets, demanding significant human efforts (Dong et al. (2023); Garg et al. (2022)). Generating the teaching signals in the time domain introduces additional computational overhead (Araki & Hattori (2023)). Researchers have attempted to adapt BP to SNNs. However, BP is inherently non-local, as it relies on computing the global error gradients. Moreover, it requires excessive memory and computational resources because the error gradients need to be stored, which contradicts the primary motivations for using SNNs, namely, high energy efficiency and low-memory computation. Furthermore, BP is incompatible with SNNs due to the non-differentiability of spiking dynamics. To mitigate these issues, the BP-based methods utilize surrogate gradients to approximate the gradients, as the non-differentiable nature of SNNs prevents direct gradient computation. The backpropagation through time (BPTT) (Werbos (2002)) can also be applied, as the original BP cannot be directly applied in the time domain. However, these approaches cause significant computational overheads and(or) lead to suboptimal convergence (Goupy et al. (2024)). Additionally, they lack biological plausibility and are incompatible with neuromorphic hardware. Given these limitations, the hitherto most effective approach for training SNNs with their energy and computational efficiency maintained is to develop a new unsupervised local learning rule based on neural behaviors that align more closely with the biological principles.

To achieve the goal above and pave the way toward brain-level efficiency in performing cognitive tasks, this paper proposes a new unsupervised local plasticity principle, which enables an SNN to perform required tasks with accuracy comparable to that of error-based backpropagation learning without external supervision. The main contributions of our paper are as follows:

- **Novel Unsupervised Plasticity Principle.** We introduce a new unsupervised learning principle, which is label-free and backward-path-free. Its unsupervised nature eliminates the need for labeled dataset, while the absence of a backward path and global error propagation, along with its local nature, substantially lowers computational costs.

- **Bio-inspired Learning Mechanism.** The proposed learning principle leverages realistic neural behaviors, including spike-based synaptic plasticity, the axonal latency model, and neurotransmitter volume transmission. Learning is driven by the timing of the spikes, which makes it well-suited to SNNs.

- **Synergistic Integration of the Biological Mechanisms.** By combining the proposed learning rules that exploits both neural connectivity and the biophysiological behavior of neurochemical components synergistically, our SNN successfully replicated the efficient learning of biological SSL. Ablation study confirms the synergistic effect of the proposed rules.

- **High-Performance Unsupervised learning.** The paper demonstrates that the proposed unsupervised learning principle achieves a record-high 100% accuracy with a human-level resolution of one degree. Applying the proposed learning principle to SNN, we successfully validated its feasibility.

## 2 RELATED WORKS

There are several bio-inspired SSL neural networks that exploit the Jeffress model-based architecture (Glackin et al. (2010); Pan et al. (2021); Zhong et al. (2022); Gao et al. (2022)) (See Section A in the Appendix for more information on the SSL and Jeffress model). The design in Glackin et al. (2010) presented the learning of SSL, mimicking the mammalian sound localization process. Based on the Jeffress model, the system has an architecture with delay elements as a surrogate for the axonal

delay pathway and the output layer. Training of the network was done by modifying the connectivity between the delay elements and the output layer using a supervised STDP with teaching signals. Before training, the delay elements and the output neurons are fully connected. The training layer, which delivers the teaching signal, connects each output neuron to a delay element. When a delay element propagates spikes to output neurons, the neuron in the training layer connecting the delay element and the output neuron relevant to the sound label provides spikes to that output neuron so that it can generate spikes. As a result, the synapse between each output neuron and one delay element will be potentiated by the STDP rule. Despite this complex training mechanism, the SSL accuracy of the system is not high.

An SSL network with high accuracy was designed by adopting multiple copies of the Jeffress model in parallel, followed by a recurrent SNN (RSNN) or a convolutional SNN (CSNN) (Pan et al. (2021)). The network has phase-coding neurons that generate the phase-locked spikes from multiple pure tones. A pair of phase-coding neurons has one set of coincidence detection neurons in each Jeffress architecture, resulting in the use of one model for processing one pure tone. The spike patterns obtained from these neurons are used as inputs to the RSNN or the CSNN. As the 2D or 3D spiking pattern of the Jeffress models are fed to RSNN or CSNN, the system is computationally costly; significantly larger than other SSL systems. In addition to its large size, it employs BPTT with surrogate gradients as the training rule, which drastically increases the computation and memory burden.

Research on building an SSL network by leveraging memristors has also been conducted. Zhong et al. (2022) proposed a network with a single-layer fully-connected SNN composed of memristor-based synapses whose learning is done by BPTT. An output neuron detecting the coincidence fires when a pair of interaural time difference (ITD)-based voltage pulses coming through memristive synapses overlap with each other. When the pulses arrive at a specific neuron simultaneously, the perfectly overlapped voltage pulses generate the largest number of spikes. When they overlap less due to the change in ITD, the number of spikes becomes lower. The resulting spike patterns of the neurons are given to the single-layer fully-connected network that classifies the azimuth of the sound source. Since the network is taught by BPTT, the learning requires heavy computation. Another design implemented an SSL using a memristor-based crossbar array (Gao et al. (2022)). The network, consisting of a single or two fully connected layers, where each synaptic weight is stored in two memristor cells, was used for the SSL. For the network learning, the backpropagation algorithm with gradient descent was applied. Nonetheless, an external processor or computer receiving the output of the memristor array to compute new weights to be updated after each mini-batch is required. This introduces a heavy computational overhead, increased latency, and degrades the learning efficiency. Furthermore, the dependency on external computation makes it less suitable for on-chip applications.

## 3 SOUND LOCALIZATION SPIKING NEURAL NETWORK WITH PROPOSED LEARNING RULES

### 3.1 OVERALL ARCHITECTURE

The architecture of sound localization SNN with initial connectivity among nucleus magnocellularis (NM) and nucleus laminaris (NL) neurons before learning, which will self-organize by the proposed plasticity rules to be described, is shown in Fig. 1a. It comprises the Jeffress model with NM-NL neurons along with the long-term plasticity (LTP) governed by the proposed learning principle, the short-term depression (STD) for more precise ITD processing (Kuba et al. (2002); Cook et al. (2003); Anwar et al. (2017)). A bunch of inhibition (INH) neurons are included to support the lateral inhibition. The superior olivary nuclei (SON) are used to compensate for the effect of ILD. The inputs to the network are pairwise spike trains, each with ITD information provided to the NM neurons on both sides, forming Jeffress model. The spike pair with ITD will propagate respectively along the axonal pathways of the NM neurons on both sides and coincide at one NL neuron having the index equal to the given ITD value, dubbed *expected winner*. The NL neuron that actually spiked the most is called *inferred winner*, indicating the estimated location of the sound source. The number of NL neurons in the network determines the sound localization resolution. For details, see Section B and Kim et al. (2025). Using the Jeffress model with arbitrary connectivity as the initial network for learning can be justified by biological evidence for the innate establishment of such a

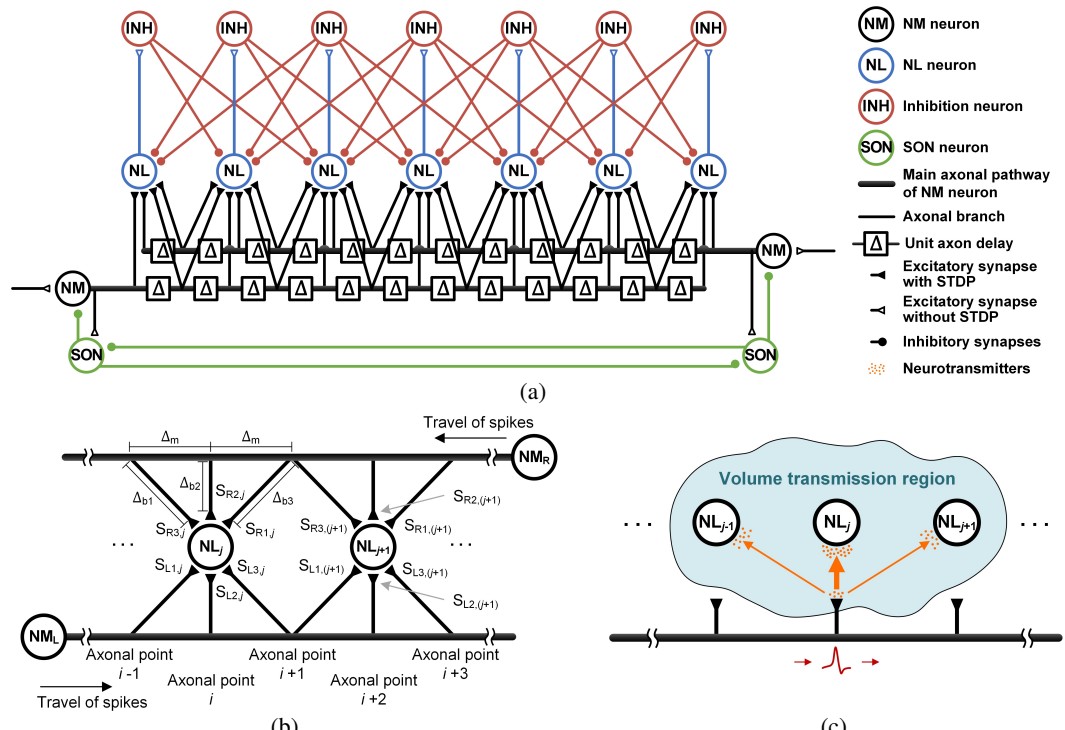

Figure 1: (a) Spiking neural network architecture for sound source localization. (b) Synaptic connectivity between the left and right NM neurons and two adjacent NL neurons. $\Delta_m$ represents the latency along the main axonal pathway, specifically between two adjacent axonal points, while $\Delta_{b1}$, $\Delta_{b2}$, and $\Delta_{b3}$ denote the latencies of the side and center axonal branches. (c) Synaptic current generation by volume transmission, which can evoke proximity learning.

network before birth (Knudsen & Knudsen (1986); Muir et al. (1989); Carr & Boudreau (1996)). The synapses between NM and NL are made learnable by the proposed learning principle, and are expected to self-organize toward the correct Jeffress model structure.

## 3.2 LATENCY-MEDIATED UNSUPERVISED STDP

In this section, a novel bio-inspired unsupervised learning rule, latency-mediated unsupervised STDP (LM-STDP), to be used to let the network learn the sound position, is proposed as a means to overcome the problems associated with supervised learning rules without performance degradation. The proposed learning rule integrates the behavioral features of the original STDP with the structural features of the auditory cortex. Combining these features enables the network to self-organize its connectivity, allowing it to perform SSL without any supervision, such as teaching signals or labels, and to evolve towards the Jeffress model from its initial connectivity.

The neuronal structure and its connectivity pattern play an important role in LM-STDP, as they determine spike propagation timing, which directly affects the weight changes governed by the rule. The axonal pathway of a neuron has a distinct propagation delay determined by structural features such as the spacing between axonal branch points and the length of the branches. The delay can influence the spike arrival time and, consequently, synaptic potentiation and depression, enabling the formation of the network with task-specific connectivity. Fig. 1b shows a segment of the Jeffress model network having connections among a few branch points in the NM neurons' axonal pathway and two NL neurons. NL neuron with index $j$, $NL_j$, shown in the center-left in Fig. 1b, has three synapses, $S_{L1,j}$, $S_{L2,j}$, and $S_{L3,j}$, connected to three axonal points, $i-1$, $i$, and $i+1$, respectively, along the main axonal pathway of the left NM neuron, $NM_L$. These multiple synapses per NL are intended to model the structure of the unlearned Jeffress model with a lot of viable synapses after hatching or birth (Knudsen & Knudsen (1986); Muir et al. (1989); Morrongiello (1988)). Due to the latency in spike propagation along the axonal pathway, a spike from the NM neuron will arrive at axonal point $i$ with a delay of $\Delta_m$ relative to its arrival at point $i-1$. As the axonal

points are assumed to be located periodically with equal distance, the latency between consecutive axonal points, $\Delta_m$, is uniform. The propagation delay of an axonal branch, $\Delta_b$, further differentiates the arrival times of spikes at different synapses. The cumulative propagation delay $\Delta_{cum}$ for each synapse, assuming the spike passes the axonal point $i-1$ at $t=0$, can be described as

$$\Delta_{cum,S_{L1,j}} = \Delta_{b1}, \ \Delta_{cum,S_{L2,j}} = \Delta_m + \Delta_{b2}, \ \Delta_{cum,S_{L3,j}} = 2\Delta_m + \Delta_{b3}, \tag{1}$$

where $\Delta_{b1} = \Delta_{b3}$ due to their symmetric connectivity. The latency of the shorter branch, $\Delta_{b2}$, is lower than those of the longer branches, $\Delta_{b1}$ and $\Delta_{b3}$. Noting that the main axonal pathway has the unit-length latency shorter than that of the axonal branch due to myelination (Susuki (2010)), the delay $\Delta_m$ is much shorter than delays $\Delta_{b1}$, $\Delta_{b2}$, and $\Delta_{b3}$, by an order of magnitude. Due to the significant latency difference between $\Delta_m$ and others, the cumulative propagation delays result in the inequality:

$$\Delta_{cum,S_{L2}} < \Delta_{cum,S_{L1}} < \Delta_{cum,S_{L3}}. \tag{2}$$

Note that, as the connectivity and the synaptic delay are identical for all NL neurons, Eq. 2 holds for every $j$ in Eq. 1. Consequently, NL neuron $j$ receives a spike from the NM neuron through three different synapses in the order of $S_2$, $S_1$, and $S_3$.

The dynamics of the synaptic weight modification by LM-STDP can be described using eligibility traces, which record the spiking history of pre- and postsynaptic neurons and determine their associated amount of synaptic weight change at any given instance of time (Morrison et al. (2008)). The dynamics of eligibility traces can be defined as

$$\frac{dx_i}{dt} = -\frac{x_i}{\tau_+} + \sum_{t_i^f} A_+ \delta(t - t_i^f)$$

$$\frac{dy_j}{dt} = -\frac{y_j}{\tau_-} - \sum_{t_j^f} A_- \delta(t - t_j^f), \tag{3}$$

where $x_i$ and $y_j$ represent the pre- and postsynaptic eligibility traces, respectively. $t_i^f$ and $t_j^f$ denote spiking times at presynaptic axonal point $i$ and postsynaptic NL neuron $j$, respectively. $A_+$ and $A_-$ refer to the scaling factors determining the magnitude of potentiation and depression, respectively. $\tau_+$ and $\tau_-$ are the time constants that govern the decay of $x_i$ and $y_j$, respectively, defining the learning window. $\delta(t - t^f)$ is the Dirac delta function that updates the eligibility traces at each spike time, $t^f$. With $x_i$ and $y_j$, the long-term synaptic weight between presynaptic axonal point $i$ and postsynaptic NL neuron $j$, governed by LM-STDP ($w_{LM-STDP_{ij}}$), can be written as

$$w_{LM-STDP_{ij}} \leftarrow w_{LM-STDP_{ij}} + \eta x_i(t) \qquad \text{at } t = t_j^f \tag{4a}$$

$$w_{LM-STDP_{ij}} \leftarrow w_{LM-STDP_{ij}} + \eta y_j(t) \qquad \text{at } t = t_i^f \tag{4b}$$

where $\eta$ is the learning rate. At presynaptic spike timing, $t_i^f$, $x_i$ increases as defined in Eq. 3 and begins to decay exponentially. If a postsynaptic spike occurs before $x_i$ decays to zero, i.e., within the learning window, the weight is potentiated at $t_j^f$ based on the value of $x_i$ at that moment, as described in Eq. 4a. In contrast, at postsynaptic spike timing, $t_j^f$, $y_j$ decreases to a negative value due to a negative $A_-$ and gradually returns to zero. If a presynaptic spike happens during this period, before $y_j$ reaches zero, the synaptic weight is weakened at $t_i^f$ according to the value of $y_j$ at $t_i^f$, as shown in Eq. 4b. Thus, a presynaptic spike before a postsynaptic spike strengthens the synapse, and a presynaptic spike that follows a postsynaptic spike weakens it. To explicitly specify the name of the synapse connected to the NL neuron, $NL_j$, the synaptic weight can be notated in terms of the synapse index. For example, the synaptic weight of $S_{L2}$ connected to the $j^{th}$ NL neuron can be denoted as $w_{LM-STDP_{L2,j}}$ instead of $w_{LM-STDP_{ij}}$ to help understanding.

With the structural characteristic of the network and the weight modification mechanism described above, LM-STDP selects only one synapse among $S_{L1,j}$, $S_{L2,j}$, and $S_{L3,j}$ to be strengthened and the others to be weakened. These synapses receive presynaptic spikes in the order of $S_{L2,j}$, $S_{L1,j}$, and $S_{L3,j}$ as defined by the order given in Eq. 2. Starting from an initial condition that the synaptic weights are sufficiently small, the $j^{th}$ NL neuron, $NL_j$, fires after receiving spikes from all three synapses. Then, all three synapses are potentiated as defined in Eq. 3 and Eq. 4a because they receive presynaptic spikes before the postsynaptic spike. After several iterations, $w_{LM-STDP_{L1,j}}$ and

$w_{LM-STDP_{L2,j}}$ will be increased enough to generate the spike of $NL_j$ earlier than the presynaptic spike of $S_{L3,j}$. When $w_{LM-STDP_{L2,j}}$ increases enough, the presynaptic spike at $S_{L2,j}$ alone can cause a spike in $NL_j$. Then, only $S_{L2,j}$ is strengthened, while the other synapses are all weakened. By repeating the procedure above, only $S_{L2,j}$, which receives the presynaptic spike first, will be fully potentiated, while others are progressively depressed and eventually deleted, resulting in the Jeffress model network connectivity. For further information and visual data, refer to Section C.

### 3.3 VOLUME TRANSMISSION-INDUCED PROXIMITY LEARNING

Along with the synaptic weight modification induced by the timing of the spikes described in the previous section, another type of the weight modification induced by the firing of adjacent neurons is proposed for accurate, efficient, and fast learning of the network. In a biological neural network, presynaptic neurons transmit their spiking information to postsynaptic neurons by releasing chemicals called neurotransmitters at synaptic clefts (Bear et al. (2016)). Since a synapse resides in a gap between neurons in the fluid-filled extracellular area, the neurotransmitter chemicals emitted can diffuse not only to the intended postsynaptic target neuron but also to adjacent neurons, which is caused by a biophysiological phenomenon called volume transmission (Agnati et al. (2010); Taber & Hurley (2014)). Local volume transmission occurs by the spillover of neurotransmitters from a synaptic cleft after its release and by the movement up to nearby clefts through the fluid-filled extracellular space. This behavior enables the membrane potentials of adjacent postsynaptic neurons to be increased along with that of the intended postsynaptic neuron.

To include the effect of this biophysiological phenomenon, we propose a neuronal proximity learning caused by the volume transmission, called volume transmission-induced proximity learning (VT-PL). The mechanism is illustrated in Fig. 1c. When an excitatory synapse connected to $NL_j$ is activated by a traveling spike, it releases neurotransmitters into the extracellular space. While most neurotransmitters are delivered to the target neuron ($NL_j$), inducing synaptic current $I_{direct,j}(t)$ to $NL_j$, a portion of neurotransmitters diffuses around the volume transmission region and may reach adjacent neurons, inducing accordingly scaled synaptic currents, $\varepsilon I_{direct,j}(t)$, to both $NL_{j-1}$ and $NL_{j+1}$. These currents are defined as volume transmission-induced synaptic currents. As a result, from the viewpoint of a specific NL neuron, $NL_j$, the synaptic current induced by collective volume transmission, $I_{indirect,j}(t)$, received from the synapses connected to $NL_{j-1}$ and $NL_{j+1}$ can be written as

$$I_{indirect,j}(t) = \varepsilon(I_{direct,j-1}(t) + I_{direct,j+1}(t)) \tag{5}$$

where $I_{direct,j-1}(t)$ and $I_{direct,j+1}(t)$ represent the excitatory synaptic currents supplied to $NL_{j-1}$ and $NL_{j+1}$ by associated synapses, respectively. $\varepsilon$ indicates the ratio between the synaptic current to the target neuron and the scaled current to an adjacent neuron. The mathematical formalization of $I_{direct}(t)$ and $I_{indirect}(t)$ can be found in Section B. As a result, the volume transmission-induced synaptic current can help increase the membrane potential of a postsynaptic neuron by repeated firing of neighboring presynaptic neurons. This interesting behavior will result in a higher firing probability of NL neurons than would otherwise be the case, thereby facilitating the efficacy of plasticity. This fact implies that it can enable adjacent NL neurons to have a higher chance of firing even when they have weak wired connections to the axonal pathway of NM neurons. The increased number of spikes will allow additional weight change by STDP. The portion of the synaptic modification by this phenomenon is similar to Eq. 4 and can be written as

$$w_{VT-PL_{ij}} \leftarrow w_{VT-PL} + \eta x_i(t) \qquad\qquad \text{at } t = t_j^f \tag{6a}$$

$$w_{VT-PL_{ij}} \leftarrow w_{VT-PL} + \eta y_j(t) \qquad\qquad \text{at } t = t_i^f \tag{6b}$$

where $\eta$ is the learning rate. This type of learning enables gradual and coordinated convergence among NL neurons, resulting in a more stable and robust learning process within the network, as supported by the evaluation results described in Section 4.

## 4 PERFORMANCE EVALUATION

To verify network behavior and performance, the SSL network with a pair of NM neurons, a pair of SON neurons, 181 NL neurons, and 181 INH neurons was designed and simulated. For the lateral inhibition, inhibitory synapses were connected to six neighboring NL neurons from each

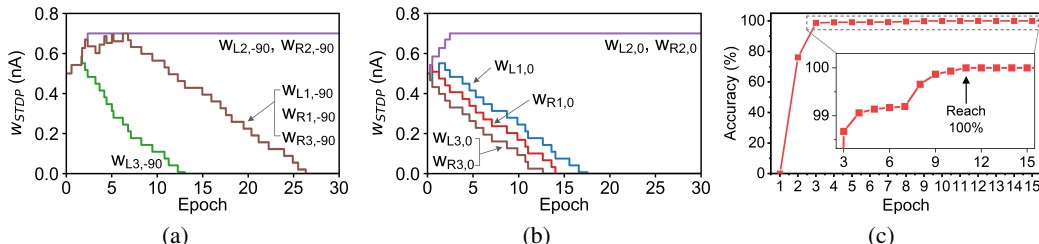

Figure 2: Migration of synaptic weights and training accuracy during learning with initial weights of 0.5 nA: (a) synaptic weights connected to $NL_{-90}$, and (b) $NL_0$. (c) SSL accuracy during learning. $w_{L1}, w_{L2}, w_{L3}, w_{R1}, w_{R2},$ and $w_{R3}$ indicate the weights of synapses $S_{L1}, S_{L2}, S_{L3}, S_{R1}, S_{R2},$ and $S_{R3}$, respectively.

INH neuron. The azimuth range of the sound source location that the system can detect is from $-90°$ to $90°$, where the 0-degree angle indicates the center direction, and the negative indexing of degrees refers to the left side. The system can distinguish the direction of the sound source with a 1-degree resolution by having 181 NL neurons that respond to input from each degree in the given azimuth range. Pairs of spikes with ITD information were used as inputs in the learning phase. Each epoch consists of presenting a total of 181 spike pairs in a random order, where each pair has an ITD corresponding to each NL neuron, which ensures that every NL neuron has opportunity to be the expected winner and change the synaptic weights connected to it. The implementation and simulation were performed by Brian2 (Goodman & Brette (2009)), an SNN simulator written in Python. The parameter values used in the simulation are listed in Table 3 in Section D.

### 4.1 PERFORMANCE OF PROPOSED LEARNING PRINCIPLE

To investigate the learning process in detail, both synaptic weight changes and overall network accuracy have been analyzed during training. Fig. 2 illustrates the migration of synaptic weight values connected to selected NL neurons during the learning process of 30 epochs and the change in the accuracy of the network. The initial weights defined in amperes are uniformly set to 0.5 nA. Fig. 2a and Fig. 2b depicts the synaptic weight changes of all six synapses ($S_{L1}, S_{L2}, S_{L3}, S_{R1}, S_{R2},$ and $S_{R3}$) in Fig. 1b connected to a single NL neuron. They indicate that the weights of synapses $S_{L2}$ and $S_{R2}$ increase continuously throughout the learning process until they reach a maximum value of 0.7 nA, while the other four synaptic weights eventually decrease to zero. After a maximum of 30 epochs of learning, the synaptic weight values are saturated at either the minimum or maximum. To assess the performance of the proposed SSL network, the accuracy was measured by observing which NL neuron fired when a single pair of spikes having a particular ITD was given. It was considered "correct" only when the inferred winner was identical to the expected winner. With uniform initial weights of 0.5 nA, the accuracy was measured 50 trials. Fig. 2c illustrates the resulting accuracy averaged over 50 trials at each epoch. Note that the inset shows a magnified view of the accuracy from the third to fifteenth epoch. The accuracy of the model exhibits a steep increase in the first few epochs, followed by a gradual convergence to 100%. This rapid learning occurs because $w_{L2}$ and $w_{R2}$, key synaptic weights in constructing the final network, increase fast during the initial epochs, as seen in Fig. 2a and Fig. 2b. As a result, the training accuracy reaches 99% after only four epochs and 100% after eleven epochs. The slow convergence after the fourth epoch is due to the gradual decrease of the synaptic weights, $w_{L1}, w_{L3}, w_{R1},$ and $w_{R3}$, connected to each NL neuron. The results demonstrate that the network, employing the proposed learning principle, efficiently learns the crucial structural features of the network and successfully attains sound localization ability.

The inference accuracy of SSL was also measured with more realistic input patterns to evaluate the performance of the network after learning. To consider more realistic environment of SSL, a pair of one-second-long input spike trains with both ITD and ILD at various firing rates and from different angles was presented to NM neurons. The firing rate of the spike trains, ranging from 1 to 500 spikes/sec, was considered to see the effect of sound intensity change. The accuracy before learning was, at best, 10% only for the centered input and 0% for all off-centered inputs. After the training with the proposed learning principle, the network achieved 100% accuracy for SSL, successfully

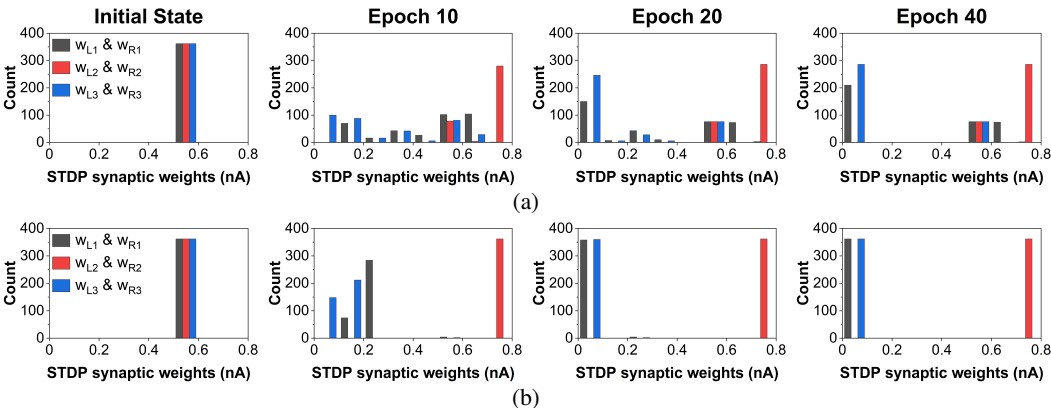

(a)

(b)

Figure 3: Distribution of synaptic weights during training (a) only with LM-STDP and (b) both with LM-STDP and VT-PL.

converging to a configuration that enables accurate one-degree resolution. This result indicates that integration of LM-STDP and VT-PL is capable of guiding SNNs to construct the optimal network.

## 4.2 ABLATION STUDY

We conduct ablation study on the proposed learning rules. Fig. 3 shows the role of VT-PL in network formation during the training process. It presents the weight distribution before and during the learning phase, with LM-STDP only and with both LM-STPD and VT-PL. The initial weight has been set to be 0.5 nA in both cases. Since the weight pairs $(w_{L1}, w_{R1})$, $(w_{L2}, w_{R2})$, and $(w_{L3}, w_{R3})$ have similar changes, as seen in Fig. 2a and Fig. 2b, they are grouped together in the histograms presented in Fig. 3. When both LM-STDP and VT-PL are applied, the

Table 1: Ablation study of learning principles.

|  | Sound source location | | |
|---|---|---|---|
|  | 0° | 45° | 88° |
| Before Training | 6% | 0% | 0% |
| + LM-STDP | 82% | 82% | 80% |
| + LM-STDP + VT-PL (ours) | **100%** | **100%** | **100%** |

synaptic weights evolve properly throughout the learning phase, leading to the formation of network architecture for effective SSL. This can be seen in the third and fourth column of Fig. 3b, where only $w_{L2}$ and $w_{R2}$ reach their maximum values, while the others decrease to near-zero values after 20 epochs, and $w_{L1}$, $w_{L3}$, $w_{R1}$, and $w_{R3}$ finish converging to zero after 40 epochs. In contrast, Fig. 3a illustrates the case with LM-STDP only. During the learning process, synaptic weights $w_{L1}$, $w_{L3}$, $w_{R1}$, and $w_{R3}$ decrease, but slowly compared to Fig. 3b. Moreover, even after 20 epochs, many synapses in all three groups still have intermediate weight values between 0.2 and 0.65 nA, which is illustrated in the third and fourth figure of Fig. 3a. In addition, there is almost no change in the weight profile between the third and fourth figure for the weight range from 0.5 to 0.65 nA, which indicates that no more learning is performed after $20^{th}$ epoch. Table 1 describes the SSL accuracy before learning, after learning only with LM-STDP, and after learning with both plasticity rules. Note that the accuracy is averaged across different input sound intensities. Because of these "unlearned" synapses, some NL neurons fail to fire, leading to a degradation in accuracy from 100% to around 80%. In summary, the network trained only with LM-STDP fails to eliminate all unnecessary connections which demonstrate that both LM-STDP and VT-PL are essential for guiding a complete synaptic reorganization, allowing the network to configure its connectivity to achieve high accuracy for the given task.

## 4.3 COMPARISON TO OTHER SSL SYSTEMS

The performance summary of the proposed and various conventional SSL networks is provided in Table 2 (Full version of Table 2 can be found in Section E). The proposed network has an identical or wider azimuth range compared to others, excluding Pan et al. (2021), and achieves the highest accuracy with the highest resolution. Compared to the computationally heavy network in Pan et al. (2021), the proposed network achieves higher resolution and accuracy with a significantly reduced

Table 2: Comparison of SLL Systems (Abridged ver.)

| Network | Learning rule | Supervised | Azimuth range | Resolution | Accuracy |
|---|---|---|---|---|---|
| Pan et al. (2021) | BPTT with surrogate gradient | ✓ | ±90° | 5° | 75.9% |
| | | | ±180° | 5° | 100%[1] |
| Glackin et al. (2010) | Supervised STDP | ✓ | ±60° | 5° | 70.63%[2] / 90.65%[3] |
| | | | | 2.5° | 78.64%[2] / 91.82%[3] |
| Zhong et al. (2022) | BPTT | ✓ | ±90° | [15°, 30°] | 96% |
| Gao et al. (2022) | Gradient descent | ✓ | ±90° | [5°, 15°] | 12.5° / 5.7°[4] |
| **This work** | **LM-STDP + VT-PL** | ✗ | ±90° | **1°** | **100%** |

network size. Unlike the network in Glackin et al. (2010), which utilizes a supervised learning method requiring labeled data, the proposed network relieves the burden by exploiting unsupervised learning. It also achieves higher accuracy with a higher resolution and a wider azimuth range while using a smaller network. Furthermore, it outperforms memristor-based SSL systems presented in Zhong et al. (2022) and Gao et al. (2022) in terms of resolution and accuracy. In conclusion, the SNN trained by the proposed unsupervised learning principle achieves highest accuracy and resolution, even though it was compared to the networks trained with the supervised learning rules.

## 5 CONCLUSION

In this paper, leveraging local plasticity and biochemical characteristics, we propose a novel unsupervised learning principle composed of latency-mediated unsupervised STDP and volume transmission-induced proximity learning for self-organizing SNNs. These unsupervised local plasticity mechanisms leverage the geometric aspects of neural connectivity and the biophysiological behavior of biochemical components, combining their effects synergistically to achieve the goal of optimally self-organizing SNNs, thereby accurately fulfilling required tasks without relying on global error backpropagation. By applying these plasticity rules, the sound source localization network successfully evolved into an architecture analogous to the Jeffress model. After a maximum of 30 epochs of learning, all learnable synaptic weights converged to either their minimum or maximum values, indicating that the network selectively potentiates task-relevant synapses while nullifying the others. The proposed SSL network achieves one-degree resolution and 100% accuracy within a realistic azimuth and ITD range. These results indicate that the proposed learning principle effectively replicates biological synaptic plasticity and self-organization, purely through neuro-inspired unsupervised learning.

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

# A  SOUND SOURCE LOCALIZATION

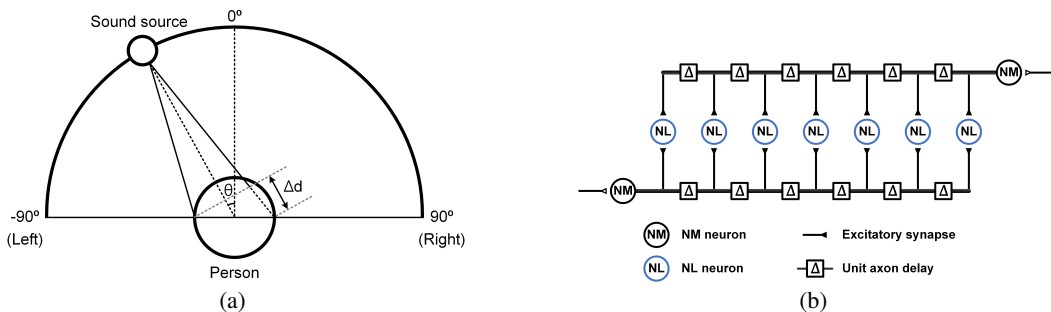

Figure 4: (a) Illustration of sound source localization (SSL) task. (b) Jeffress model.

Sound source localization (SSL) is a task determining the localization of a sound source. Fig. 4a describes SSL task. Jeffress model shown in Fig. 4b illustrates the avian SSL mechanism. The NM neuron on each side receives spikes from the cochlea. Each NL neuron functions as a coincidence detector by integrating spikes from the ipsilateral and contralateral NMs. The axonal pathway from each NM neuron, stretching across the entire network, causes a propagation delay to each NL neuron. The architecture enables NL neurons to receive coincident spikes from both sides under specific ITD conditions. For example, when the sound source is located straight ahead, the sound will reach both NM neurons simultaneously and let the NL neuron at the center fire more than the others. When the sound source deviates from the straight-ahead direction, the sound arrives at both NM neurons at different times, creating an ITD. Then, the NM spikes generated by the earlier sound start propagating earlier along the delay line, resulting in the firing of the NL neuron at an off-centered position. In this way, the NL neurons in the Jeffress model can identify the locations of sound sources.

# B  MODELING DETAILS OF THE NEURON AND SYNAPSE

The STD employed in each synapse among NM and NL neurons helps the NL neuron to have a relatively constant spiking rate regardless of the sound intensity. It also minimizes the NL neurons' spiking rate, making the network more energy efficient. The lateral inhibition applied among NL neurons through INH neurons prevents neighboring NL neurons from firing after one neuron fires. It helps distinguish a single inferred winner from the others. Each INH neuron receives excitatory current from the associated NL neuron and supplies inhibitory current to neighboring NL neurons. SON neurons connected with respective NM neurons will inhibit NM neurons from firing to maintain their firing rate relatively constant and, more importantly, to reduce the disturbance by the ILD.

In our sound source localization network, shown in Fig. 1a, the synapses connecting NM-NL have a major role, in which the long-term weights are governed by the proposed learning principle, described in Section 3. On top of it, STD plasticity is also involved to provide an advantage of accurate ITD processing (Kim et al. (2025)). The STD employed in each synapse among NM and NL neurons helps the NL neuron to have a relatively constant spiking rate regardless of the sound intensity. It also minimizes the NL neurons' spiking rate, making the network more energy efficient. The STD part of the synapse can be described as

$$\frac{dw_{STD}(t)}{dt} = -\frac{1}{0.0613} w_{STD}(t) + z(t) - \sum_k \delta(t - t_k)(1 - D)w_{STD}(t_k^-) \tag{7a}$$

$$\frac{dz(t)}{dt} = -\frac{1}{0.040} w_{STD}(t) + \frac{1}{0.040} \tag{7b}$$

where $t_k$ indicates the timing of presynaptic spikes and $z(t)$ is an internal variable in a pair of first-order ODEs governing the short-term plasticity behavior (Kim et al. (2025)). Note that the measurement data from the rat was used to get the coefficients in Eq. 7 to use the biological STD data (Jia et al. (2004)). The resulting total effective synaptic weight, considering the LTP and STD simultaneously, can be written as

$$W_{eff}(t) = w_{LTP}(t)w_{STD}(t). \tag{8}$$

In this equation, $w_{LTP}(t)$ indicates the long-term weight modifications governed by LM-STDP and VT-PL as given in Eq. 4 and Eq. 6, respectively. Note that $w_{LTP}(t)$ in our network is not a linear summation of $w_{LM-STDP}(t)$ and $w_{VT-PL}(t)$ because these effects are combined as excitatory synaptic currents and the inhibitory synaptic current is also involved to form the total synaptic current, as will be seen below. $w_{STD}(t)$ is given by Eq. 7.

The alpha synaptic current model (Weisstein; De Schutter (2009)), describing the biological excitatory postsynaptic current (EPSC), was employed to accurately model the synaptic current dynamics in the network. The alpha synaptic current supplied to the $j^{th}$ NL neuron directly from its own synapses can be described as

$$I_{direct,j}(t) = \sum_i \sum_p u(t - t_{i,p}) W_{eff}(t_{i,p}^-) \cdot \left( \frac{t - t_{i,p}}{\tau_\alpha} \right) e^{1 - \frac{t - t_{i,p}}{\tau_\alpha}}, \tag{9}$$

where $I_{direct,j}(t)$ denotes the direct synaptic current to the $j^{th}$ NL neuron. $t_{i,p}$ is the $p^{th}$ presynaptic spike timing, the time when spike arrives at axonal point $i$. The first sigma over $i$ represents the summation of all the axonal points with synaptic connections to the $j^{th}$ NL neuron. The volume transmission-induced indirect synaptic current, $I_{indirect,j}(t)$, shown in Eq. 5 can be similarly written to the expression in Eq. 9, resulting in the total excitatory synaptic current to $NL_j$, $I_{exc,j}(t)$, written as

$$I_{exc,j}(t) = I_{direct,j}(t) + I_{indirect,j}(t). \tag{10}$$

On top of the excitatory synaptic current, the lateral inhibition supplies the inhibitory synaptic current to NL neurons. The lateral inhibition applied among NL neurons through INH neurons prevents neighboring NL neurons from firing after one neuron fires. It helps distinguish a single inferred winner from the others. Each INH neuron receives excitatory current from the associated NL neuron and supplies inhibitory current to neighboring NL neurons. SON neurons connected with respective NM neurons will inhibit NM neurons from firing to maintain their firing rate relatively constant and, more importantly, to reduce the disturbance by the ILD. The amount of inhibitory synaptic current that the $j^{th}$ NL neuron receives can be written as

$$I_{inh,j}(t) = W_{inh} \sum_s u(t - t_{s,j}) \cdot \left( \frac{t - t_{s,j}}{\tau_\alpha} \right) e^{1 - \frac{t - t_{s,j}}{\tau_\alpha}}, \tag{11}$$

where $t_{s,j}$ indicates the $s^{th}$ spike timing of the $j^{th}$ inhibitory neuron, which is the presynaptic spike at the inhibitory synapse, and $W_{inh}$ represents the fixed weight of the inhibitory synapse. The total synaptic current to $j^{th}$ NL neuron is then defined as the sum of the excitatory and inhibitory synaptic currents as

$$I_{total,j}(t) = I_{exc,j}(t) + I_{inh,j}(t), \tag{12}$$

where $I_{exc,j}(t)$ and $I_{inh,j}(t)$ represent the total excitatory and inhibitory synaptic currents, as defined in Eq. 10 and Eq. 11, respectively.

For implementing the biological neuron in the proposed neural network, the LIF neuron model (Gerstner et al. (2014)) is used for all the neurons in the network. By the LIF model, the membrane potential of a neuron is defined as

$$\tau_m \frac{dV_m(t)}{dt} = -(V_m(t) - V_{rest}) + R_m I_{total}(t), \tag{13}$$

where $V_m$ is the membrane potential, $\tau_m$ (= $R_m C_m$) is the membrane time constant, $I_{total}(t)$ is the total input synaptic current defined in Eq. 12, and $V_{rest}$ is the resting potential. Whenever the membrane potential reaches the threshold voltage, $V_{th}$, the neuron fires, and the membrane potential resets to the reset voltage, $V_{reset}$, during the refractory period.

The learning behavior of a few NL neurons adjacent to each other in the network, based on the proposed plasticity principle, is depicted in Fig. 5 by simulated waveforms of the synaptic current, membrane potential, and weight migration of the six synapses connected to each NL neuron. The waveforms show the synaptic current and membrane potential of NL neuron indexes 0 and 1, as well as the weight value of the synapses connected to each of them when a single spike pair having ITD=0 ms is given, letting $NL_0$ be the expected winner. The first column shows the waveforms of $NL_0$, where the spikes from NM neurons coincide, and the second column shows those of the

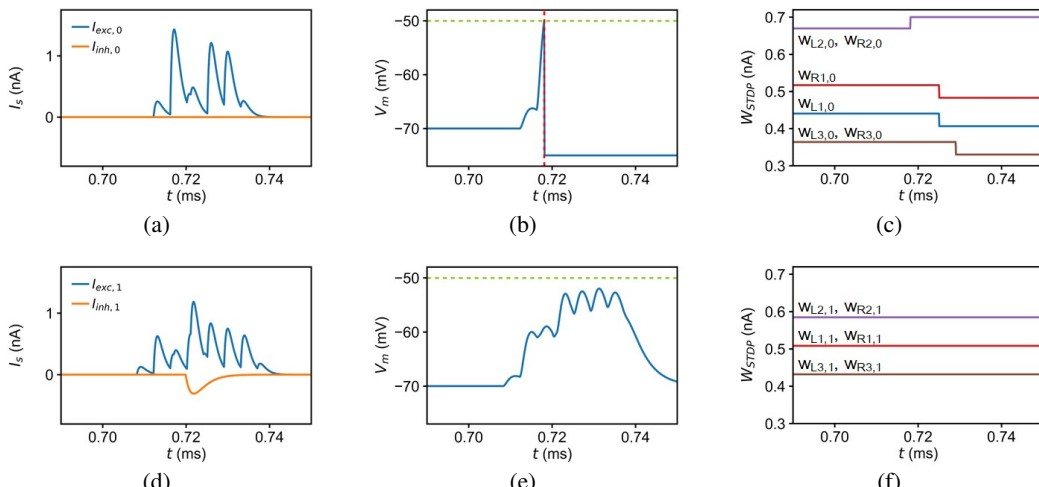

Figure 5: Weight modification by LM-STDP and VT-PL when ITD=0 ms. (a) and (d) synaptic currents to $NL_0$ and $NL_1$. (b) and (e) membrane potentials of $NL_0$ and $NL_1$. (c) and (f) synaptic weights connected to $NL_0$ and $NL_1$. The first row shows waveforms of $NL_0$ and the second row shows $NL_1$.

neighboring neuron, $NL_1$. As three spike pairs coincide at $NL_0$, it receives excitatory synaptic currents of high amplitudes three times through the synapses, as illustrated in Fig. 5a. Note that each peak indicates the coincidence of a spike pair from the left and right NM neurons, and as there are three pairs of synapses connected to one NL neuron, three peaks of $I_{direct,j}(t)$ occur in Fig. 5a. Due to this synaptic current having a high amplitude, the membrane potential of $NL_0$ increases rapidly and reaches the threshold, as shown in Fig. 5b. As the first synaptic current pair arrives before 0.72 ms, causing the postsynaptic spike at $NL_0$, the weights of these two synapses, $w_{L2,0}$ and $w_{R2,0}$, increase, and the weights of the other synapses decrease, as depicted in Fig. 5c. The volume transmission-induced synaptic current $I_{indirect,j}(t)$ is also visible in Fig. 5a and Fig. 5d, with small peaks other than three high peaks in Fig. 5a and additional small peaks other than six peaks in Fig. 5d. In $NL_1$, although the membrane potential increases to near $V_{th}$ with the help of VT-PL, it fails to fire because of the inhibitory synaptic current caused by the spike of $NL_0$, as shown in Fig. 5e. The long-term synaptic weights, $w_{LTP}(t)$, of the synapses connected to $NL_1$ did not change in Fig. 5f, as the postsynaptic spike did not occur in Fig. 5e.

## C  WEIGHT UPDATE IN LM-STDP

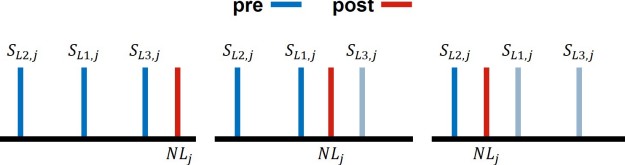

Figure 6: Timing sequence of pre- and postsynaptic spikes at synapses $S_{L1,j}$, $S_{L2,j}$, and $S_{L3,j}$ around the $j^{th}$ NL neuron, $NL_j$.

As seen in Fig. 1a, each NL neuron receives three synaptic inputs from an NM neuron, following the connectivity in Fig. 1b. Given that there are two NM neurons, one on each side, three pairs of synaptic currents delivered through three pairs of synapses will coincide at the expected winner. As depicted in Fig. 6, when the initial synaptic weight values are small, the expected winner will fire only after receiving all three pairs of spikes. As the synaptic weights increase, even the first spike pair can deliver enough synaptic current for the expected winner to fire. By the LM-STDP rule, only the pair of synapses transmitting the earliest spikes will be potentiated, while others will be weakened. In addition, the volume transmission described in Fig. 1c enables the proximity learning mechanism, facilitating and coordinating stable and robust network learning. By combining these

mechanisms, the proposed learning principle successfully guides the entire network to strengthen the correct synapses and weaken the others, thereby forming the connectivity of the Jeffress model.

## D  NETWORK PARAMETERS

Table 3: Parameter values for network simulation.

| Parameter | Value |
|---|---|
| Membrane resting potential | −70 mV |
| Membrane reset potential | −75 mV |
| Firing threshold | −50 mV |
| Refractory period | 0.5 ms |
| Membrane resistance of the NM neuron | 40 MΩ |
| Membrane capacitance of the NM neuron | 60 pF |
| Membrane resistance of the SON neuron | 40 MΩ |
| Membrane capacitance of the SON neuron | 60 pF |
| Membrane resistance of the NL neuron | 40 MΩ |
| Membrane capacitance of the NL neuron | 0.1 pF |
| Membrane resistance of the inhibition neuron | 40 MΩ |
| Membrane capacitance of the inhibition neuron | 0.25 pF |
| Unit axon delay ($\Delta$) | 2 us |
| STDP weight increase factor ($A_+$) | 0.85 nA |
| STDP weight decrease factor ($A_-$) | −0.6824 nA |
| Time constant for the STDP curve at $\Delta t > 0$ ($\tau_+$) | 2 ms |
| Time constant for the STDP curve at $\Delta t < 0$ ($\tau_-$) | 2 ms |
| Learning rate ($\eta$) | 0.5 |
| Ratio of $I_{indirect}$ to $I_{direct}$ ($\varepsilon$) | 0.2 |
| Depression rate ($D$) | 0.40506 |
| Time constant for the recovery of STD ($\tau_{STD1}$) | 16.4 ms |
| Time constant for the recovery of STD ($\tau_{STD2}$) | 2469.7 ms |
| Time constant for the alpha synaptic current ($\tau_\alpha$) | 0.001-0.5 ms |
| Initial long-term synaptic weight ($w_{LTP}$) | 0.5 nA |
| Final long-term synaptic weight ($w_{LTP}$) | 0-0.7 nA |

## E  COMPARISON OF SSL SYSTEMS

Table 4: Comparison of SLL Systems (Full ver.)

| Network | Modeling strategy | Learning rule | Supervised | Input stimulus | Max. ITD (ms) | Frequency range (Hz) |
|---|---|---|---|---|---|---|
| Pan et al. (2021) | Biological model | BPTT with surrogate gradient | ✓ | Speech | [±0.31, ±1.88] | [200, 800] |
| Glackin et al. (2010) | Biological model | Supervised STDP | ✓ | Recorded pure tone | ±0.5 | [600, 1,600] |
| Zhong et al. (2022) | ANN with memristors | BPTT | ✓ | Paired pulse | ±0.0005 | N/A |
| Gao et al. (2022) | ANN with memristive array | Gradient descent | ✓ | Recorded sound pulse | N/A | N/A |
| **Ours** | **Biological model** | **LM-STDP + VT-PL** | ✗ | Paired pure tone | ±0.72 | [1, 500] |

Table 4: Comparison of SLL Systems (Full ver.) (cont.)

| Network | Frequency range (Hz) | Azimuth range | # of neurons | # of synapses | Resolution | Accuracy |
|---|---|---|---|---|---|---|
| Pan et al. (2021) | [200, 800] | $\pm 90°$ | 27,176 / 39,176[5] | >1,715,200 / 754,580 | 5° | 75.9% |
| | | $\pm 180°$ | | | | 100%[1] |
| Glackin et al. (2010) | [600, 1,600] | $\pm 60°$ | 525 | 7,665 | 5° | 70.63%[2] / 90.65%[3] |
| | | | 1,029 | 27,321 | 2.5° | 78.64%[2] / 91.82%[3] |
| Zhong et al. (2022) | N/A | $\pm 90°$ | 35 | 250 | [15, 30°] | 96% |
| Gao et al. (2022) | N/A | $\pm 90°$ | 67 / 261 | 420 / 10,550 | [5°, 15°] | 12.5° / 5.7°[4] |
| **Ours** | [1, 500] | $\pm 90°$ | 366 | 2,359[6] / 1,635[7] | **1°** | **100%** |

---

[1] Four microphones are used for sound capturing

[2] Accuracy based on spikes within $\pm 5°$ of the target azimuth angle

[3] Accuracy based on spikes within $\pm 10°$ of the target azimuth angle

[4] The Angle deviation (angular difference between inference result and sound source position) in RMS using single-/two-layer networks

[5] Neurons in RSNN and CSNN, respectively

[6] Number of synapses before learning

[7] Number of synapses after learning

