# OpenReview forum: "A Bio-Inspired Sound Localization Spiking Neural Network with Unsupervised Local Plasticity and Proximity Learning"
_ICLR.cc/2026/Conference — Submitted to ICLR 2026_

### Official Review · Reviewer_pspf · 2025-10-27

**Soundness:** 2
**Presentation:** 2
**Contribution:** 2
**Rating:** 2
**Confidence:** 3

**Summary:**

This paper presents an unsupervised learning principle for spiking neural networks (SNNs) that aims to achieve biologically plausible self-organization without relying on global error backpropagation. The proposed mechanism integrates two bio-inspired rules: latency-mediated spike-timing-dependent plasticity (LM-STDP), which incorporates axonal delays into synaptic updates, and volume transmission-induced proximity learning (VT-PL), which models neurotransmitter diffusion among neighboring neurons. The method is evaluated on a sound source localization (SSL) task, demonstrating accurate one-degree localization and synaptic organization similar to the Jeffress model.

**Strengths:**

1. The emphasis on unsupervised local plasticity rules is valuable for both neuroscience modeling and neuromorphic computing.

2. The work has clear biological grounding. It references established auditory neuroscience models and data, including Jeffress, short-term depression mechanisms, and neurophysiological constants.

**Weaknesses:**

1. While LM-STDP and VT-PL are each inspired by biological phenomena, the mechanistic rationale for their combination is not clearly established. The interaction between latency effects and volume transmission remains heuristic rather than analytically or experimentally justified. It is unclear whether the synergy arises naturally or was tuned for this specific task.

2. The evaluation is confined to a single SSL task. It remains uncertain whether the proposed learning rule can scale to more complex tasks, modalities, or architectures

3. Although Table 2 compares accuracy and resolution with earlier SSL works, it does not include recent SNN frameworks using surrogate gradients or biologically constrained BPTT variants. Additionally, it is unclear why the proposed method is better than BPTT.

4. The paper reports that all weights converge to binary extremes (0 or maximum) after 30 epochs. This all-or-none behavior may indicate an overly aggressive learning dynamic and could compromise representational flexibility or robustness under noisy inputs.

5. Real auditory signals involve noise and non-stationarity. The current experiments lack robustness evaluations.

**Questions:**

1. What theoretical or empirical evidence supports combining LM-STDP and VT-PL? Could one rule alone suffice for partial learning?

2. Can this dual-rule learning principle be extended to more complex SSL tasks?

3. How does the proposed training compare computationally to BPTT?

4. How does the model behave with noisy inputs, and does learning still converge stably?

5. How sensitive are the results to key parameters?

---

> ### Author Response · Authors · 2025-12-03
>
> We thank the reviewer for the comments. We would like to offer the following clarifications.
>
> >_**Weakness 1**. While LM-STDP and VT-PL are each inspired by biological phenomena, the mechanistic rationale for their combination is not clearly established. The interaction between latency effects and volume transmission remains heuristic rather than analytically or experimentally justified. It is unclear whether the synergy arises naturally or was tuned for this specific task._
>
> >_**Question 1**. What theoretical or empirical evidence supports combining LM-STDP and VT-PL? Could one rule alone suffice for partial learning?_
>
> Although LM-STDP is the one that directly changes synaptic weights, VT-PL is a crucial component for achieving high accuracy and stability. Without LM-STDP, applying VT-PL alone would lead to learning failure, resulting in a significant drop in accuracy to less than 10%, comparable to the “Before Training” baseline in Table 1. While VT-PL can technically modify synaptic weights as described in Eq. (6), it lacks temporal information, as the axonal delay defining the specific spike arrival timing is a component of LM-STDP. Consequently, without LM-STDP, VT-PL would amplify random firing activities, rather than forming an efficient structure for the task.
>
> Relying solely on LM-STDP also has limitations. As shown in our ablation study (Table 1 in Section 4.2), training with only LM-STDP results in incomplete learning, achieving an accuracy of only around 81%. Applying VT-PL on top of LM-STDP to resolve the accuracy drop helps the learning process in two main aspects:
>
> -	Ensuring complete structural learning by avoiding “silent neurons”
>
>   Let us first consider training with only LM-STDP. When an NL neuron that would normally fire may fail to fire at its corresponding ITD due to its small synaptic weights, a neighboring NL neuron can fire instead. Then, the latter triggers lateral inhibition on the former. This causes negative feedback; the non-firing neuron cannot change its weights since the postsynaptic spike is missing, and the neighbor’s strengthened synapses make the firing neuron have a higher probability to fire for the next input spikes. Consequently, the synapses connected to the non-firing NL neuron can remain “unlearned”, stuck at the initial weights as seen in Fig. 3, letting the neuron become a “silent neuron” as it will never fire. The degree assigned to “silent” NL neurons cannot be recognized, which leads to the accuracy drop shown in Table 1.
>
>   If VT-PL is included in the training process, it momentarily increases the firing probability of the neighboring NL neurons by providing an indirect excitatory synaptic current (Eq. (5)). This lets the neurons that initially fall behind the learning process also be given a chance to spike and participate in the LM-STDP learning process. This cooperation prevents NL neurons from becoming “silent,” enabling complete convergence of all synapses and achieving 100% accuracy.
>
>
>
> -	Enhancing robustness to noise
>
>     Applying VT-PL increases the robustness of the proposed learning principle. When the noise is applied to the system, it can cause an incorrect NL neuron to spike due to an incorrect ITD. This non-ideal spike incorrectly strengthens synapses that are not ideal (e.g., $S_{L1}$ or $S_{L3}$ in Fig. 1(b)). This leads to an increase in multiple synaptic weights (e.g., $w_{L1}$, $w_{L2}$, $w_{L3}$, and $w_{R1}$ in Fig. 1(b) all increase), failing to construct the precise structure. VT-PL mitigates this by potentiating neighborhoods. Each NL neuron only receives a substantial amount of current when the given ITD corresponds to itself or to neighboring neurons. This prevents NL neurons from firing to random ITDs, thereby mitigating the strengthening of incorrect synapses. This guides the network toward the correct structure even under the noisy conditions. To provide empirical evaluation for the robustness claims above, we have conducted extended simulations. The simulation result includes the training phase using two different learning principles, “only LM-STDP” and “LM-STDP and VT-PL”, under various noise levels. The result demonstrates that training with “LM-STDP and VT-PL” ensures the stable convergence to ideal values regardless of noise intensity, while training with “only LM-STDP” suffers from the synaptic weights remaining at the intermediate values and failed to converge under high noise level. In conclusion, this result has proven the necessity of VT-PL for noise robustness, as it facilitates neighboring neurons to fire and prevents the neuron to become silent. The simulation result can be found here: https://figshare.com/s/4528f4adc66eb559af02.
>
> This confirms that the synergy of both proposed learning rules are inseparable from a structured network to achieve high accuracy.

---

> ### Author Response · Authors · 2025-12-03
>
> ________________________________________
>
> >_**Weakness 2**. The evaluation is confined to a single SSL task. It remains uncertain whether the proposed learning rule can scale to more complex tasks, modalities, or architectures_
>
> >_**Question 2**. Can this dual-rule learning principle be extended to more complex SSL tasks?_
>
> The proposed learning principle can be extended to more complex SSL tasks. By increasing the number of NL neurons in the network, the system can distinguish smaller degree differences in sound source location, achieving higher resolution. For example, increasing the number of neurons by a factor of 2 enables the network to achieve 0.5-degree resolution for SSL. We are currently training the larger network with more neurons. The simulation results proving the learning capability of the proposed learning principle on a larger network will be submitted as soon as they are available.
>
> Moreover, the proposed principle is not limited to the SSL task. It is a principle that can be effectively applied to the self-organization of structured SNNs, in which network configuration and connectivity play important roles in performing cognitive functions. Some other potential examples are as follows:
>
> -	Modeling Grid cells in the entorhinal cortex, and Place cells that receive signals from grid cells. They enable animals to navigate a space and to self-localize by integrating their movement and the environment [1]-[4]. The grid cells form a hexagonal structure that requires a precise configuration, which our proposed principle can effectively establish.
>
> -	Implementing the Ellipsoid Body (EB) in the Central Complex (CX) of insects, which plays an important role in spatial orientation, visual memory, and path navigation [5], [6].
>
> These applications are not only about biomimicry but can also be expanded to real-world applications. In robotics, the above applications can be used for simultaneous localization and mapping (SLAM) [7]. They can also be applied to computer vision [8], even in virtual reality (VR) systems. Our principle provides the unsupervised, low-power learning capabilities required for the on-chip implementation of these structured neural networks.
>
> We will include the above discussion in the revised version of the paper.
>
> ________________________________________
> >_**Weakness 3**. Although Table 2 compares accuracy and resolution with earlier SSL works, it does not include recent SNN frameworks using surrogate gradients or biologically constrained BPTT variants. Additionally, it is unclear why the proposed method is better than BPTT._
>
> >_**Question 3**. How does the proposed training compare computationally to BPTT?_
>
>
> Our approach demonstrates strong advantages over BPTT-based learning [9]. Since our learning principle relies on unsupervised local learning, it circumvents the excessive memory overhead of storing all forward states and backward gradients, as well as the high computational complexity of BPTT. This is critical because BPTT compromises the SNN’s core advantage of high energy efficiency and low-memory computation, and it hinders the real-world application of SNNs with neuromorphic hardware.
>
> -	Memory complexity
>    As BPTT requires storing all the data during the T time step for all N neurons, it requires $O(N\cdot T)$ memory. On the other hand, our unsupervised local learning principle requires memory of $O(N)$ to store the current state of each neuron.
>
> -	Computational complexity
>
>    While the complexity for BPTT and our learning principles may appear similar, both scaling linearly with time and neurons, $O(N\cdot T)$, the actual computational burden differs fundamentally in two aspects:
>
>
>   - Computation type
>
>     One computation in BPTT involves hardware-intensive operations, such as matrix multiplication and division for normalization. In contrast, our learning principle relies on sparse, event-driven accumulation. Updating a neuron’s state, the membrane potential, is a simple addition operation that corresponds to the intrinsic dynamics of neuromorphic hardware, requiring significantly less power and area.
>
>
>   - Absence of the backward pass
>
>     While our method operates solely in the forward pass, updating weights locally and instantly, BPTT requires a backward pass to propagate errors, doubling the number of operations per time step. Therefore, even with the same complexity $O(N\cdot T)$, our method has a much smaller constant factor.
>
> In conclusion, our unsupervised local learning principle is more efficient for training SNNs while maintaining their inherent strengths, energy, and computational efficiency.

---

> ### Author Response · Authors · 2025-12-03
>
> ________________________________________
>
> >_**Weakness 4**. The paper reports that all weights converge to binary extremes (0 or maximum) after 30 epochs. This all-or-none behavior may indicate an overly aggressive learning dynamic and could compromise representational flexibility or robustness under noisy inputs._
>
> We would like to clarify that convergence to binary extremes does not indicate overly aggressive learning dynamics, but rather reflects our structural selection process. In spatio-temporal tasks like sound source localization, the network's final structure itself is a form of “representation”. If synapses remain at intermediate weights, the neuron becomes sensitive to a broad range of latencies, leading to temporal blurring and low resolution, or it may become a “silent neuron” that does not respond to any input, leading to a drop in accuracy. The binary convergence indicates that the network has successfully performed synaptic pruning, eliminating all ambiguous pathways and retaining only the single precise delay line required for the task.
>
> Binary extreme weights also enhance the robustness of the network. A neuron with many intermediate weights is prone to firing due to accumulated random noise, leading to spurious firing for inaccurate inputs. However, by setting the weights of the irrelevant synapses to zero, the network ignores those inaccurate input timings. Simultaneously, the relevant synapses with the highest synaptic weights ensure that the neuron fires only when the precise coincidence occurs. As a result, the network with binary extreme weights can achieve 100% accuracy as described in Section 4.1, specifically in Fig. 2.
>
> ________________________________________
>
> >_**Weakness 5**. Real auditory signals involve noise and non-stationarity. The current experiments lack robustness evaluations_
>
>
> >_**Question 4**. How does the model behave with noisy inputs, and does learning still converge stably?_
>
> When the noise is applied to the system, it can cause an incorrect NL neuron to spike due to an incorrect ITD. This non-ideal spike incorrectly strengthens synapses that are not ideal (e.g., $S_{L1}$ or $S_{L3}$ in Fig. 1(b)). This leads to an increase in multiple synaptic weights (e.g., $w_{L1}$, $w_{L2}$, $w_{L3}$, and $w_{R1}$ in Fig. 1(b) all increase), failing to construct the precise structure. VT-PL mitigates this by potentiating neighborhoods. Each NL neuron only receives a substantial amount of current when the given ITD corresponds to itself or to neighboring neurons. This prevents NL neurons from firing to random ITDs, thereby mitigating the strengthening of incorrect synapses. This guides the network toward the correct structure even under the noisy conditions. To provide empirical evaluation for the robustness claims above, we have conducted extended simulations. The simulation result includes the training phase using two different learning principles, “only LM-STDP” and “LM-STDP and VT-PL”, under various noise levels. The result demonstrates that training with “LM-STDP and VT-PL” ensures the stable convergence to ideal values regardless of noise intensity, while training with “only LM-STDP” suffers from the synaptic weights remaining at the intermediate values and failed to converge under high noise level. In conclusion, this result has proven the necessity of VT-PL for noise robustness, as it facilitates neighboring neurons to fire and prevents the neuron to become silent. The simulation result can be found here: https://figshare.com/s/4528f4adc66eb559af02.
>
> ________________________________________
>
> >_**Question 5**. How sensitive are the results to key parameters?_
>
> Most key parameters listed in Table 3 were directly adapted from biological data of the avian auditory cortex [10]. Also, we have validated the model’s robustness by testing it across a wide range of input intensities with fixed parameters. As described in Section 4.1, the network achieved 100% accuracy across input firing rates ranging from 1 to 500 spikes/sec.
> Also, standard SNNs can be sensitive to parameter values, such as thresholds and learning rates, often leading to “silent neurons” when settings are not precise. However, our VT-PL mechanism reduces the chance of a neuron becoming “silent”, as described in the first comment, comment on Weakness 1 and Question 1. By providing additional synaptic current to neighboring neurons, VT-PL prevents neurons from becoming silent even when local excitation is insufficient. This mechanism widens the safe operating range of parameters, ensuring convergence without “precise” fine-tuning.

---

> ### Author Response · Authors · 2025-12-03
>
> ________________________________________
>
> [Reference]
>
> [1] Hafting, T., Fyhn, M., Molden, S., Moser, M. B., & Moser, E. I. (2005). Microstructure of a spatial map in the entorhinal cortex. Nature, 436(7052), 801-806.
>
> [2] Burak, Y., & Fiete, I. R. (2009). Accurate path integration in continuous attractor network models of grid cells. PLoS computational biology, 5(2), e1000291.
>
> [3] Park, S. W., Jang, H. J., Kim, M., & Kwag, J. (2019). Spatiotemporally random and diverse grid cell spike patterns contribute to the transformation of grid cell to place cell in a neural network model. PloS one, 14(11), e0225100.
>
> [4] O'Keefe, J., & Dostrovsky, J. (1971). The hippocampus as a spatial map: preliminary evidence from unit activity in the freely-moving rat. Brain research.
>
> [5] Seelig, J. D., & Jayaraman, V. (2015). Neural dynamics for landmark orientation and angular path integration. Nature, 521(7551), 186-191.
>
> [6] Turner-Evans, D., Wegener, S., Rouault, H., Franconville, R., Wolff, T., Seelig, J. D., ... & Jayaraman, V. (2017). Angular velocity integration in a fly heading circuit. Elife, 6, e23496.
>
> [7] Milford, M. J., Wyeth, G. F., & Prasser, D. (2004, April). RatSLAM: a hippocampal model for simultaneous localization and mapping. In IEEE International Conference on Robotics and Automation, 2004. Proceedings. ICRA'04. 2004 (Vol. 1, pp. 403-408). IEEE.
>
> [8] Liu, J., Xu, W., Li, X., & Zheng, X. (2021). Improved Visual Recognition Memory Model Based on Grid Cells for Face Recognition. Frontiers in Neuroscience, 15, 718541.
>
> [9] Knapp, C., & Carter, G. (2003). The generalized correlation method for estimation of time delay. IEEE transactions on acoustics, speech, and signal processing, 24(4), 320-327.
>
> [10] Kim, J., Zhou, P., Wi, U., Joo, B., Choi, D., Seol, M. L., ... & Kong, B. S. (2024). Biomimetic Spiking Neural Network Based on Monolayer 2-D Synapse With Short-Term Plasticity for Auditory Brainstem Processing. IEEE Transactions on Cognitive and Developmental Systems, 17(2), 247-258.

---

### Official Review · Reviewer_7Nrx · 2025-10-30

**Soundness:** 2
**Presentation:** 3
**Contribution:** 1
**Rating:** 2
**Confidence:** 4

**Summary:**

This paper proposes a SNN-based method for high-precision SSL. The backbone of the proposed SNN is similar to the conventional Jeffress model that leverages the sound signal arrival disparity between left and right receivers depending on the sound source location. The STDP learner can fine-tune the disparity between the two receivers, which allows higher angle resolutions. Further, the proposed proximity learning can encourage the neurons with low synaptic weights, which otherwise likely turn into dead neurons. As a result, the proposed method achieves higher accuracy of SSL than several emerging methods.

**Strengths:**

1.	Technical novelty. Although the learners used are renowned, the application of these learners to SSL tasks is novel.
2.	High performance: The proposed method achieves higher performance than “emerging” methods.

**Weaknesses:**

1.	Limited scientific novelty: As such, this method is based on two well-known physiological rules (STDP and proximity learning), which are just used to Jeffress model-like SNNs.
2.	Limited impact: This method is only for SSL, which has very limited impact. The present manuscript does not address any other application domains of larger impact.
3.	Weak performance evaluation: The technical evaluation of the proposed method is weak, which includes accuracy only. There are a number of aspects that should be considered, like complexity, wallclock time, parallelism, and so forth. Further, this work should be compared with the conventional works along with emerging works.

**Questions:**

What are technical advantages of this work over the Jeffres model?

---

> ### Author Response · Authors · 2025-12-03
>
> We thank the reviewer for the time in reviewing our paper.
>
> >_**Weakness 1**. Limited scientific novelty: As such, this method is based on two well-known physiological rules (STDP and proximity learning), which are just used to Jeffress model-like SNNs._
>
> The scientific novelty of our paper lies in the synergistic integration of LM-STDP and VT-PL, yielding a new unified local unsupervised learning principle. Our work is NOT a simple application of the well-known learning rules. We propose and prove that the combination of these two bio-inspired synaptic mechanisms is essential for the efficient and stable learning of structured spiking neural networks (SNNs) such as the Jeffress model. LM-STDP uses the latency in the axonal propagation pathway to select the most efficient synapses and discard the others. However, training with only LM-STDP leads to incomplete learning, as shown in Table 1 and Fig. 3 of the ablation study in Section 4.2, leaving redundant synapses and causing accuracy degradation. The proposed VT-PL, inspired by the volume transmission of the neurotransmitter shown in Fig. 1(c) in biological synaptic transmission, increases the probability of firing of the NL neurons near spike coincidence. It helps neurons with small synaptic weights fire so that their synaptic weights can change via LM-STDP. Their synergistic behavior achieves complete and stable learning successfully, thereby increasing the SSL accuracy to 100%. This result proves that a novel solution to overcome the instability of the unsupervised STDP.
>
> To help the better understanding, we also explain the effect of applying VT-PL. VT-PL is a crucial component for achieving high accuracy and stability. It helps the learning process in two main aspects:
>
> -	Ensuring complete structural learning by avoiding “silent neurons”
>
>    As shown in the Ablation study (Section 4.2, Fig. 3, and Table 1), relying solely on LM-STDP results in incomplete learning, yielding an accuracy of around 80%. (Note that, with both LM-STDP and VT-PL, the accuracy goes up to nearly 100%.)
>
>    Let us first consider training with only LM-STDP. When an NL neuron that would normally fire may fail to fire at its corresponding ITD due to its small synaptic weights, a neighboring NL neuron can fire instead. Then, the latter triggers lateral inhibition on the former. This causes negative feedback; the non-firing neuron cannot change its weights since the postsynaptic spike is missing, and the neighbor’s strengthened synapses make the firing neuron have a higher probability to fire for the next input spikes. Consequently, the synapses connected to the non-firing NL neuron can remain “unlearned”, stuck at the initial weights as seen in Fig. 3, letting the neuron become a “silent neuron” as it will never fire. The degree assigned to “silent” NL neurons cannot be recognized, which leads to the accuracy drop shown in Table 1.
>
>    If VT-PL is included in the training process, it momentarily increases the firing probability of the neighboring NL neurons by providing an indirect excitatory synaptic current (Eq. (5)). This lets the neurons that initially fall behind the learning process also be given a chance to spike and participate in the LM-STDP learning process. This cooperation prevents NL neurons from becoming “silent,” enabling complete convergence of all synapses and achieving 100% accuracy.
>
> (To be continued)

---

> ### Author Response · Authors · 2025-12-03
>
> (continued)
>
> -   Enhancing robustness to noise
>
>     Applying VT-PL increases the robustness of the proposed learning principle. When the noise is applied to the system, it can cause an incorrect NL neuron to spike due to an incorrect ITD. This non-ideal spike incorrectly strengthens synapses that are not ideal (e.g., $S_{L1}$ or $S_{L3}$ in Fig. 1(b)). This leads to an increase in multiple synaptic weights (e.g., $w_{L1}$, $w_{L2}$, $w_{L3}$, and $w_{R1}$ in Fig. 1(b) all increase), failing to construct the precise structure. VT-PL mitigates this by potentiating neighborhoods. Each NL neuron only receives a substantial amount of current when the given ITD corresponds to itself or to neighboring neurons. This prevents NL neurons from firing to random ITDs, thereby mitigating the strengthening of incorrect synapses. This guides the network toward the correct structure even under the noisy conditions. To provide empirical evaluation for the robustness claims above, we have conducted extended simulations. The simulation result includes the training phase using two different learning principles, “only LM-STDP” and “LM-STDP and VT-PL”, under various noise levels. The result demonstrates that training with “LM-STDP and VT-PL” ensures the stable convergence to ideal values regardless of noise intensity, while training with “only LM-STDP” suffers from the synaptic weights remaining at the intermediate values and failed to converge under high noise level. In conclusion, this result has proven the necessity of VT-PL for noise robustness, as it facilitates neighboring neurons to fire and prevents the neuron to become silent. The simulation result can be found here: https://figshare.com/s/4528f4adc66eb559af02.
>
>
> The synergistic combination of LM-STDP and VT-PL is therefore essential for robust and precise self-organization. We will include the above discussion in the revised version of the paper.
>
> ______________________________________

---

> ### Author Response · Authors · 2025-12-03
>
> >_**Weakness 2**. Limited impact: This method is only for SSL, which has very limited impact. The present manuscript does not address any other application domains of larger impact._
>
>
> The proposed principle is not limited to the SSL task. It is a principle that can be effectively applied to the self-organization of structured SNNs, in which network configuration and connectivity play important roles in performing cognitive functions. Some other potential examples are as follows:
>
>
>   -	Modeling Grid cells in the entorhinal cortex, and Place cells that receive signals from grid cells. They enable animals to navigate a space and to self-localize by integrating their movement and the environment [1]-[4]. The grid cells form a hexagonal structure that requires a precise configuration, which our proposed principle can effectively establish.
>
>
>   -	Implementing the Ellipsoid Body (EB) in the Central Complex (CX) of insects, which plays an important role in spatial orientation, visual memory, and path navigation [5], [6].
>
> These applications are not only about biomimicry but can also be expanded to real-world applications. In robotics, the above applications can be used for simultaneous localization and mapping (SLAM) [7]. They can also be applied to computer vision [8], even in virtual reality (VR) systems. Our principle provides the unsupervised, low-power learning capabilities required for the on-chip implementation of these structured neural networks. We will add this generalization of our principle in the discussion.

---

> ### Author Response · Authors · 2025-12-03
>
> >_**Weakness 3**. weak performsance evaluation: The technical evaluation of the proposed method is weak, which includes accuracy only. There are a number of aspects that should be considered, like complexity, wallclock time, parallelism, and so forth. Further, this work should be compared with the conventional works along with emerging works._
>
>
> We provide a comparison of our learning principle with three distinct approaches: conventional sound source localization (SSL), standard artificial neural networks (ANNs), and supervised spiking neural networks (SNNs) with BPTT. While all-clock time in software simulations depends on the platform (CPU/GPU architecture) and platform-specific software optimizations (e.g. FFT libraries), it does not reflect the true latency or efficiency on neuromorphic hardware. Therefore, we have conducted a comparative analysis focusing on computational complexity, memory complexity, parallelism and the target hardware, which are crucial for the hardware implementation. Table R1 summarizes the comparison between our proposed principle and conventional/emerging methods, and the comparison items, including ‘computational complexity and optimal hardware’, ‘memory complexity’, and ‘parallelism’, are elaborated further below.
>
>
>
> Table R1. Comparison of complexity and hardware efficiency
> |  | Conventional DSP (GCC-PHAT) | Standard ANN | Supervised SNN with BPTT | **Proposed** |
> | :--- | :--- | :--- | :--- | :--- |
> | **Optimal HW platform** | CPU | GPU or AI accelerator | GPU or AI accelerator | **Neuromorphic hardware** |
> | **Dominant Operation** | Complex Multiplication (FFT) | Matrix Multiplication | Matrix Multiplication | **Sparse accumulation** |
> | **Computational complexity** | $O(M \log M)$ | $O(N^2)$ per layer | $O(N\cdot T)$ + Backward path | **$O(N\cdot T)$, without Backward path** |
> | **Memory complexity** | $O(M)$ | High ($O(W)$) | Very high ($O(N\cdot T)$) | **Low ($O(N)$)** |
> | **Parallelism** | Limited (sequential FFT steps) | High | High | **High** |
> | **Learning** | N/A | Supervised | Supervised | **Unsupervised** |
>
> > * $N$: Number of neurons, $T$: Time steps, $M$: Number of samples in the signal window (DSP)
>
>
> A detailed explanation of each comparison metric presented in Table R1 is provided below. The Table R1 and the regarding discussion will be included in the revised version of the paper.
>
> - Computational complexity and the optimal hardware
>
>   Conventional digital signal processing (DSP) methods [9], [10] rely on computationally intensive operations such as Fourier transforms (FFT) and cross-correlation. These methods involve extensive arithmetic operations like multiplication of complex numbers, requiring CPU or dedicated DSP chips. Note that since DSP methods require M data points to start computation, their computational complexity is $O(M log M)$. Standard ANNs [11] and supervised SNNs [12] rely on backpropagation that is computationally and memory-intensive. These methods depend highly on dense matrix multiplication, demanding the use of GPUs, which consumes large amount of power. In contrast, our proposed method relies on LIF neuron dynamics, in which a neuron’s state is updated by the simple and sparse accumulation of synaptic currents. Updating a neuron’s state, the membrane potential, involves simple addition operation that corresponds to the intrinsic dynamics of neuromorphic hardware, requiring significantly less power and area. While our method operates solely in the forward pass, updating weights locally and instantly, BPTT requires a backward pass to propagate errors, doubling the number of operations per time step. Therefore, even with the theoretical complexity $O(N\cdot T)$, our method has a much smaller constant factor.
>
>
> - Memory complexity
>
>   Our method also offers a critical advantage in terms of memory efficiency. Standard ANNs require to store all the weights and activations for forward and backward computation. Supervised SNNs trained with BPTT must store membrane potentials and synaptic currents across all time steps (T) for all N neurons to compute gradients, resulting in a memory complexity of $O(N\cdot T)$. On the other hand, our unsupervised, local learning principle requires $O(N)$  memory to store the current state of each neuron.
>
>
> - Parallelism
>
>   Unlike DSP algorithms that require sequential processing steps (e.g., FFT), our proposed method is inherently parallel. Since our learning principle is local, every neuron and synapse can update its state (e.g., membrane potential, synaptic weights) independently. This characteristic ensures massive parallelism when implemented on neuromorphic chip.
>
> (To be continued)

---

> > ### Author Response · Authors · 2025-12-03
> >
> > (continued)
> >
> > For the last sentence in the reviewer’s comment, asking for a comparison with conventional works along with emerging works, the answer is as follows. The systems using ANN or BPTT methods are included in the comparison tables in the manuscript (Table 2 and Table 4). Gao et al [13] relied on gradient descent and used a significantly larger number of synapses, yet achieved lower resolution than ours. Pan et al. [12] used the BPTT method and built a much larger network than ours, but only achieved 5-degree resolution. The other systems in Table 2 exhibit computational inefficiencies similar to ours. This implies that our approach is highly efficient compared to others, achieving superior resolution with reduced overhead.
> >
> > The above discussion and Table R1 will be included in the revised version of the paper.

---

> ### Author Response · Authors · 2025-12-03
>
> >_**Question**. What are technical advantages of this work over the Jeffres model?_
>
> It is important to clarify that our contribution is not the implementation of the Jeffress model, but rather the proposal of a novel unsupervised local learning principle that enables the self-organization of structural SNNs. In this context, the Jeffress model is the result of our learning process, rather than an opponent we are competing with.
>
> The Jeffress model has fixed, predetermined delay lines and structure, and it lacks adaptability. However, our approach, a synergistic combination of LM-STDP and VT-PL, allows the SNN to learn the optimal connection from an arbitrary initial state. This adaptability provides an advantage, particularly regarding potential neuromorphic hardware implementation. In physical hardware, device mismatches or PVT (process, voltage, and temperature) variations can degrade the performance of the fixed model. In contrast to the static Jeffress model, our approach can allow the system to calibrate itself by learning optimal connections based on effective delays, thereby ensuring robustness against physical variations.
>
> ________________________________________
> [Reference]
>
> [1] Hafting, T., Fyhn, M., Molden, S., Moser, M. B., & Moser, E. I. (2005). Microstructure of a spatial map in the entorhinal cortex. Nature, 436(7052), 801-806.
>
> [2] Burak, Y., & Fiete, I. R. (2009). Accurate path integration in continuous attractor network models of grid cells. PLoS computational biology, 5(2), e1000291.
>
> [3] Park, S. W., Jang, H. J., Kim, M., & Kwag, J. (2019). Spatiotemporally random and diverse grid cell spike patterns contribute to the transformation of grid cell to place cell in a neural network model. PloS one, 14(11), e0225100.
>
> [4] O'Keefe, J., & Dostrovsky, J. (1971). The hippocampus as a spatial map: preliminary evidence from unit activity in the freely-moving rat. Brain research.
>
> [5] Seelig, J. D., & Jayaraman, V. (2015). Neural dynamics for landmark orientation and angular path integration. Nature, 521(7551), 186-191.
>
> [6] Turner-Evans, D., Wegener, S., Rouault, H., Franconville, R., Wolff, T., Seelig, J. D., ... & Jayaraman, V. (2017). Angular velocity integration in a fly heading circuit. Elife, 6, e23496.
>
> [7] Milford, M. J., Wyeth, G. F., & Prasser, D. (2004, April). RatSLAM: a hippocampal model for simultaneous localization and mapping. In IEEE International Conference on Robotics and Automation, 2004. Proceedings. ICRA'04. 2004 (Vol. 1, pp. 403-408). IEEE.
>
> [8] Liu, J., Xu, W., Li, X., & Zheng, X. (2021). Improved Visual Recognition Memory Model Based on Grid Cells for Face Recognition. Frontiers in Neuroscience, 15, 718541.
>
> [9] Knapp, C., & Carter, G. (2003). The generalized correlation method for estimation of time delay. IEEE transactions on acoustics, speech, and signal processing, 24(4), 320-327.
>
> [10] GCC-PHAT with Speech-oriented Attention for Robotic Sound Source Localization
>
> [11] Chakrabarty, S., & Habets, E. A. (2017, October). Broadband DOA estimation using convolutional neural networks trained with noise signals. In 2017 IEEE workshop on applications of signal processing to audio and acoustics (WASPAA) (pp. 136-140). IEEE.
>
> [12] Pan, Z., Zhang, M., Wu, J., Wang, J., & Li, H. (2021). Multi-tone phase coding of interaural time difference for sound source localization with spiking neural networks. IEEE/ACM Transactions on Audio, Speech, and Language Processing, 29, 2656-2670.
>
> [13] Gao, B., Zhou, Y., Zhang, Q., Zhang, S., Yao, P., Xi, Y., ... & Wu, H. (2022). Memristor-based analogue computing for brain-inspired sound localization with in situ training. Nature communications, 13(1), 2026.

---

### Official Review · Reviewer_C6N5 · 2025-10-31

**Soundness:** 3
**Presentation:** 4
**Contribution:** 4
**Rating:** 8
**Confidence:** 4

**Summary:**

The paper introduces a biologically inspired, unsupervised learning principle for spiking neural networks that combines
- a latency-mediated STDP (LM-STDP) which explicitly leverages axonal propagation delays,
- and a volume-transmission proximity learning (VT-PL) mechanism that models neurotransmitter spillover to neighboring neurons.

**Strengths:**

The results are impressive, demonstrating 100% accuracy with a 1-degree resolution, outperforming all four supervised counterparts used for comparison. The biological plausibility and unsupervised nature make it highly relevant for energy-efficient, on-chip applications.

**Weaknesses:**

The paper is well written, with clear biological motivation. Yet, it is unclear if the learning process is entirely unsupervised, since I read in line 342: "The results are impressive, demonstrating 100% accuracy with a 1-degree resolution, outperforming many supervised counterparts. The biological plausibility and unsupervised nature make it highly relevant for energy-efficient, on-chip applications." This implies that during training, the system knows which NL neuron should fire (the "expected winner") for a given ITD. This is a form of weak supervision or a structured input paradigm.

The paper reports a perfect accuracy, it is unclear however how the proposed model would perform under realistic acoustic settings with noise, reverberation. Besides, I would recommend to add a discussion on scalability and generalisation, since the network is evaluated on a very specific task with synthetic spike pairs with controlled ITD/ILD and firing rates.

**Questions:**

What is the exact of supervision? This should ABSOLUTELY be clarified if the paper is accepted.

Could you discuss the robustness, scalability, and generalisation capabilities of the presented model?

---

> ### Author Response · Authors · 2025-12-03
>
> We appreciate the positive assessment of the paper and the suggestions for improvement. Our answer to your questions and concerns is provided below.
>
> >_**Weakness (1st item)**. The paper is well written, with clear biological motivation. Yet, it is unclear if the learning process is entirely unsupervised, since I read in line 342: "The results are impressive, demonstrating 100% accuracy with a 1-degree resolution, outperforming many supervised counterparts. The biological plausibility and unsupervised nature make it highly relevant for energy-efficient, on-chip applications." This implies that during training, the system knows which NL neuron should fire (the "expected winner") for a given ITD. This is a form of weak supervision or a structured input paradigm._
>
> >_**Question (1st item)**. What is the exact of supervision? This should ABSOLUTELY be clarified if the paper is accepted._
>
> The learning process is strictly unsupervised. The ground-truth label (the correct location of the sound source) of the given data is NOT provided to the system. LM-STDP, mainly described in Eq. (3) and (4), and VT-PL, described in Eq. (5) and (6), clearly do not include any form of supervision signal such as “error” or “target”. LM-STDP updates the synaptic weights directly following Eq. (4), which is governed only by pre- or postsynaptic spike timing. The term “expected winner” is used only in the post-training evaluation to define the ground-truth neuron for the given spike train needed to calculate the sound source localization accuracy.
>
> We agree that the input is structured data that contains ITD information. This is a characteristic of the SSL task, in which the input (ITD) is directly linked to the output (the neural index or location). However, using a structured input does not necessarily mean supervision of the learning process.
>
> In conclusion, our approach is fully unsupervised as the network successfully self-organizes its structure without requiring any external labels, backward error paths, or even a teaching signal.

---

> ### Author Response · Authors · 2025-12-03
>
> >_**Weakness (2nd item)**. The paper reports a perfect accuracy, it is unclear however how the proposed model would perform under realistic acoustic settings with noise, reverberation. Besides, I would recommend to add a discussion on scalability and generalisation, since the network is evaluated on a very specific task with synthetic spike pairs with controlled ITD/ILD and firing rates._
>
> >_**Question (2nd item)**. Could you discuss the robustness, scalability, and generalisation capabilities of the presented model?_
>
> -	Generalization capability
>
>    The proposed principle is not limited to the SSL task. It is a principle that can be effectively applied to the self-organization of structured SNNs, in which network configuration and connectivity play important roles in performing cognitive functions. Some other potential examples are as follows:
>
>
>   -	Modeling Grid cells in the entorhinal cortex, and Place cells that receive signals from grid cells. They enable animals to navigate a space and to self-localize by integrating their movement and the environment [1]-[4]. The grid cells form a hexagonal structure that requires a precise configuration, which our proposed principle can effectively establish.
>
>
>   -	Implementing the Ellipsoid Body (EB) in the Central Complex (CX) of insects, which plays an important role in spatial orientation, visual memory, and path navigation [5], [6].
>
>
>    These applications are not only about biomimicry but can also be expanded to real-world applications. In robotics, the above applications can be used for simultaneous localization and mapping (SLAM) [7]. They can also be applied to computer vision [8], even in virtual reality (VR) systems. Our principle provides the unsupervised, low-power learning capabilities required for the on-chip implementation of these structured neural networks.
>
>
>
> -  Robustness to noise
>
>    When the noise is applied to the system, it can cause an incorrect NL neuron to spike due to an incorrect ITD. This non-ideal spike incorrectly strengthens synapses that are not ideal (e.g., $S_{L1}$ or $S_{L3}$ in Fig. 1(b)). This leads to an increase in multiple synaptic weights (e.g., $w_{L1}$, $w_{L2}$, $w_{L3}$, and $w_{R1}$ in Fig. 1(b) all increase), failing to construct the precise structure. VT-PL mitigates this by potentiating neighborhoods. Each NL neuron only receives a substantial amount of current when the given ITD corresponds to itself or to neighboring neurons. This prevents NL neurons from firing to random ITDs, thereby mitigating the strengthening of incorrect synapses. This guides the network toward the correct structure even under the noisy conditions. To provide empirical evaluation for the robustness claims above, we have conducted extended simulations. The simulation result includes the training phase using two different learning principles, “only LM-STDP” and “LM-STDP and VT-PL”, under various noise levels. The result demonstrates that training with “LM-STDP and VT-PL” ensures the stable convergence to ideal values regardless of noise intensity, while training with “only LM-STDP” suffers from the synaptic weights remaining at the intermediate values and failed to converge under high noise level. In conclusion, this result has proven the necessity of VT-PL for noise robustness, as it facilitates neighboring neurons to fire and prevents the neuron to become silent. The simulation result can be found here: https://figshare.com/s/4528f4adc66eb559af02.
>
>
>
>
>
> -  Scalability of the network
>
>    The proposed architecture is inherently scalable due to its local characteristics. The resolution depends on the number of NL neurons covering the ITD range. Increasing the number of NL neurons, e.g., from 181 to 362, to achieve a finer resolution of 0.5° does not require changing the learning principle or the SNN architecture. It is important to note that the computational complexity of our model scales linearly with the number of neurons ($O(N)$). This linear complexity ensures feasibility for larger-scale implementation of our network.
>
> The above discussions will be included in the revised version of the paper.

---

> ### Author Response · Authors · 2025-12-03
>
> [Reference]
>
> [1] Hafting, T., Fyhn, M., Molden, S., Moser, M. B., & Moser, E. I. (2005). Microstructure of a spatial map in the entorhinal cortex. Nature, 436(7052), 801-806.
>
> [2] Burak, Y., & Fiete, I. R. (2009). Accurate path integration in continuous attractor network models of grid cells. PLoS computational biology, 5(2), e1000291.
>
> [3] Park, S. W., Jang, H. J., Kim, M., & Kwag, J. (2019). Spatiotemporally random and diverse grid cell spike patterns contribute to the transformation of grid cell to place cell in a neural network model. PloS one, 14(11), e0225100.
>
> [4] O'Keefe, J., & Dostrovsky, J. (1971). The hippocampus as a spatial map: preliminary evidence from unit activity in the freely-moving rat. Brain research.
>
> [5] Seelig, J. D., & Jayaraman, V. (2015). Neural dynamics for landmark orientation and angular path integration. Nature, 521(7551), 186-191.
>
> [6] Turner-Evans, D., Wegener, S., Rouault, H., Franconville, R., Wolff, T., Seelig, J. D., ... & Jayaraman, V. (2017). Angular velocity integration in a fly heading circuit. Elife, 6, e23496.
>
> [7] Milford, M. J., Wyeth, G. F., & Prasser, D. (2004, April). RatSLAM: a hippocampal model for simultaneous localization and mapping. In IEEE International Conference on Robotics and Automation, 2004. Proceedings. ICRA'04. 2004 (Vol. 1, pp. 403-408). IEEE.
>
> [8] Liu, J., Xu, W., Li, X., & Zheng, X. (2021). Improved Visual Recognition Memory Model Based on Grid Cells for Face Recognition. Frontiers in Neuroscience, 15, 718541.

---

### Official Review · Reviewer_Wcen · 2025-10-31

**Soundness:** 2
**Presentation:** 2
**Contribution:** 2
**Rating:** 2
**Confidence:** 3

**Summary:**

This paper aims to provide a bio-inspired solution to sound location via two main ideas: a latency-mediated spike timing-dependent plasticity rule, and the volume transmission of neurotransmitters.

**Strengths:**

**Pros**:
- Being biologically inspired and plausible makes this work pretty interesting;
- The proposed solution is unsupervised, and offers good performance for sound localization.

**Weaknesses:**

**Cons**:

- One of the two key ideas, a latency-mediated spike timing-dependent plasticity rule, is not fully clearly described. The authors should provide a better motivation and a deeper insight of this learning rule. It seems to be designed specifically for sound localization based on the Jeffress model.

- The other idea,  use of the volume transmission of neurotransmitters, is a simple twist. It amounts to redirecting part of the total postsynaptic current to nearby neurons. How is going to improve the performance? Any insights?

- The application space of the proposed techniques is pretty narrow; they seem to work only for one application and one model (Jeffress model); The authors should discuss the broader generality of their approach.

**Questions:**

- While both proposed ideas are somewhat biologically inspired, their relevance and insights regarding the learning process are not missing. The authors should provide more in-depth discussions to justify their design choices.

---

> ### Author Response · Authors · 2025-12-03
>
> We thank the reviewer for the comments. We would like to discuss your concerns.
>
> > _**Weakness (1st item)**. One of the two key ideas, a latency-mediated spike timing-dependent plasticity rule, is not fully clearly described. The authors should provide a better motivation and a deeper insight of this learning rule. It seems to be designed specifically for sound localization based on the Jeffress model._
>
> > _**Question**. While both proposed ideas are somewhat biologically inspired, their relevance and insights regarding the learning process are not missing. The authors should provide more in-depth discussions to justify their design choices._
>
>
> While backpropagation-based methods achieve high accuracy, they have excessive computational overhead and require labeled datasets. Conversely, the original unsupervised STDP, the brain’s basic learning rule, offers advantages in energy and computational efficiency. However, due to the lack of a global error correction mechanism, it often results in low accuracy. Furthermore, most existing SNN approaches rely on off-chip training (e.g., using CPU and GPU) followed by inference-only deployment, limiting their real-time adaptability. To fully exploit the inherent energy efficiency of SNNs and enable their wider application, a robust and accurate learning rule is essential. We propose a novel learning principle that achieves high accuracy while maintaining high energy efficiency.
>
> Inspired by how the biological brain resolves the tradeoff between efficiency and accuracy, our goal is to simulate the biological phenomenon of fine network formation, learned postnatally through experience. The formation of the sound localization network evolves through two phases. First, the network structure that propagates signals from NM to NL is established innately. This corresponds to our network’s initial architecture, having all the neurons but with unrefined connectivity. Second, after birth, the precise synaptic connectivity is refined through experience with auditory stimuli [1]-[3]. The motivation for LM-STDP is to model the “experience-dependent refinement” of the sound source localization network. We provide the network with coarse connectivity and use LM-STDP to select and refine the delay lines based on the input sounds, exactly as observed in the biological brain.
>
> The fundamental insight of LM-STDP is that it is a structural selection mechanism that transforms temporal information into spatial connectivity. This aligns with the findings by Gerstner et al. [4], who demonstrated that an unsupervised Hebbian rule can resolve the “temporal paradox” by selectively reinforcing the connections with matching delays. As described in Fig. 1(b), the main axonal pathway corresponds to myelinated axons having fast transmission, while the axonal branches have slower transmission speeds. This structural geometry creates a specific sequence of spike arrival times. If standard STDP were used without accounting for these specific latencies, the synapse receiving the earliest arriving spike due to the short distance to NM (e.g., S_L1 and S_R1 in Fig. 1(b)) would be wrongly potentiated regardless of whether it contributed to coincidence, because early arriving spikes have a higher chance to meet the “pre-before-post” spike condition. This prevents the formation of a Jeffress model architecture and leads to incorrect coincidence detection, as synapses would be potentiated regardless of the ITD. Our LM-STDP solves this problem by strictly correlating the spike arrival time with the postsynaptic spike. It potentiates only the specific branches that delivered the spike precisely, thereby pruning the “too fast” or “too slow” synapses. This ensures the formation of the correct Jeffress model architecture.

---

> ### Author Response · Authors · 2025-12-03
>
> > _**Weakness (2nd item)**. The other idea, use of the volume transmission of neurotransmitters, is a simple twist. It amounts to redirecting part of the total postsynaptic current to nearby neurons. How is going to improve the performance? Any insights?_
>
> > _**Question**. While both proposed ideas are somewhat biologically inspired, their relevance and insights regarding the learning process are not missing. The authors should provide more in-depth discussions to justify their design choices._
>
> VT-PL is a crucial component for achieving high accuracy and stability. It helps the learning process in two main aspects:
>
> -  Ensuring complete structural learning by avoiding “silent neurons”
>
>     As shown in the Ablation study (Section 4.2, Fig. 3, and Table 1), relying solely on LM-STDP results in incomplete learning, yielding an accuracy of around 80%. (Note that, with both LM-STDP and VT-PL, the accuracy goes up to nearly 100%.)
>
>     Let us first consider training with only LM-STDP. When an NL neuron that would normally fire may fail to fire at its corresponding ITD due to its small synaptic weights, a neighboring NL neuron can fire instead. Then, the latter triggers lateral inhibition on the former. This causes negative feedback; the non-firing neuron cannot change its weights since the postsynaptic spike is missing, and the neighbor’s strengthened synapses make the firing neuron have a higher probability to fire for the next input spikes. Consequently, the synapses connected to the non-firing NL neuron can remain “unlearned”, stuck at the initial weights as seen in Fig. 3, letting the neuron become a “silent neuron” as it will never fire. The degree assigned to “silent” NL neurons cannot be recognized, which leads to the accuracy drop shown in Table 1.
>
>     If VT-PL is included in the training process, it momentarily increases the firing probability of the neighboring NL neurons by providing an indirect excitatory synaptic current (Eq. (5)). This lets the neurons that initially fall behind the learning process also be given a chance to spike and participate in the LM-STDP learning process. This cooperation prevents NL neurons from becoming “silent,” enabling complete convergence of all synapses and achieving 100% accuracy.
>
>
> -  Enhancing robustness to noise
>
>     Applying VT-PL increases the robustness of the proposed learning principle. When the noise is applied to the system, it can cause an incorrect NL neuron to spike due to an incorrect ITD. This non-ideal spike incorrectly strengthens synapses that are not ideal (e.g., $S_{L1}$ or $S_{L3}$ in Fig. 1(b)). This leads to an increase in multiple synaptic weights (e.g., $w_{L1}$, $w_{L2}$, $w_{L3}$, and $w_{R1}$ in Fig. 1(b) all increase), failing to construct the precise structure. VT-PL mitigates this by potentiating neighborhoods. Each NL neuron only receives a substantial amount of current when the given ITD corresponds to itself or to neighboring neurons. This prevents NL neurons from firing to random ITDs, thereby mitigating the strengthening of incorrect synapses. This guides the network toward the correct structure even under the noisy conditions. To provide empirical evaluation for the robustness claims above, we have conducted extended simulations. The simulation result includes the training phase using two different learning principles, “only LM-STDP” and “LM-STDP and VT-PL”, under various noise levels. The result demonstrates that training with “LM-STDP and VT-PL” ensures the stable convergence to ideal values regardless of noise intensity, while training with “only LM-STDP” suffers from the synaptic weights remaining at the intermediate values and failed to converge under high noise level. In conclusion, this result has proven the necessity of VT-PL for noise robustness, as it facilitates neighboring neurons to fire and prevents the neuron to become silent. The simulation result can be found here: https://figshare.com/s/4528f4adc66eb559af02.
>
> The synergistic combination of LM-STDP and VT-PL is therefore essential for robust and precise self-organization. We will include the above discussion in the revised version of the paper.

---

> ### Author Response · Authors · 2025-12-03
>
> > _**Weakness (3rd item)**. The application space of the proposed techniques is pretty narrow; they seem to work only for one application and one model (Jeffress model); The authors should discuss the broader generality of their approach._
>
> The proposed principle is not limited to the SSL task. It is a principle that can be effectively applied to the self-organization of structured SNNs, in which network configuration and connectivity play important roles in performing cognitive functions. Some other potential examples are as follows:
>
> - Modeling Grid cells in the entorhinal cortex, and Place cells that receive signals from grid cells. They enable animals to navigate a space and to self-localize by integrating their movement and the environment [5]-[8]. The grid cells form a hexagonal structure that requires a precise configuration, which our proposed principle can effectively establish.
>
> - Implementing the Ellipsoid Body (EB) in the Central Complex (CX) of insects, which plays an important role in spatial orientation, visual memory, and path navigation [9], [10].
>
> These applications are not only about biomimicry but can also be expanded to real-world applications. In robotics, the above applications can be used for simultaneous localization and mapping (SLAM) [11]. They can also be applied to computer vision [12], even in virtual reality (VR) systems. Our principle provides the unsupervised, low-power learning capabilities required for the on-chip implementation of these structured neural networks. We will include the above discussion in the revised version of the paper.

---

> ### Author Response · Authors · 2025-12-03
>
> [Reference]
>
> [1] D. W. Muir, R. K. Clifton, and M. G. Clarkson, “The Development of a Human Auditory Localization Response: A U-Shaped Function,” Can. J. Psychol., vol. 43, no. 2, pp. 199–216, Jun. 1989, doi: 10.1037/h0084220.
>
> [2] E. Kezuka, S. Amano, and V. Reddy, “Developmental Changes in Locating Voice and Sound in Space,” Front. Psychol., vol. 8, Sep. 2017, doi: 10.3389/fpsyg.2017.01574.
>
> [3] B. A. Morrongiello, “Infants’ Localization of Sounds Along the Horizontal Axis: Estimates of Minimum Audible Angle,” Dev. Psychol., vol. 24, no. 1, pp. 8–13, Jan. 1988, doi: 10.1037/0012-1649.24.1.8.
>
> [4] Gerstner, W., Kempter, R., Van Hemmen, J. L., & Wagner, H. (1996). A neuronal learning rule for sub-millisecond temporal coding. Nature, 383(6595), 76-78.
>
> [5] Hafting, T., Fyhn, M., Molden, S., Moser, M. B., & Moser, E. I. (2005). Microstructure of a spatial map in the entorhinal cortex. Nature, 436(7052), 801-806.
>
> [6] Burak, Y., & Fiete, I. R. (2009). Accurate path integration in continuous attractor network models of grid cells. PLoS computational biology, 5(2), e1000291.
>
> [7] Park, S. W., Jang, H. J., Kim, M., & Kwag, J. (2019). Spatiotemporally random and diverse grid cell spike patterns contribute to the transformation of grid cell to place cell in a neural network model. PloS one, 14(11), e0225100.
>
> [8] O'Keefe, J., & Dostrovsky, J. (1971). The hippocampus as a spatial map: preliminary evidence from unit activity in the freely-moving rat. Brain research.
>
> [9] Seelig, J. D., & Jayaraman, V. (2015). Neural dynamics for landmark orientation and angular path integration. Nature, 521(7551), 186-191.
>
> [10] Turner-Evans, D., Wegener, S., Rouault, H., Franconville, R., Wolff, T., Seelig, J. D., ... & Jayaraman, V. (2017). Angular velocity integration in a fly heading circuit. Elife, 6, e23496.
>
> [11] Milford, M. J., Wyeth, G. F., & Prasser, D. (2004, April). RatSLAM: a hippocampal model for simultaneous localization and mapping. In IEEE International Conference on Robotics and Automation, 2004. Proceedings. ICRA'04. 2004 (Vol. 1, pp. 403-408). IEEE.
>
> [12] Liu, J., Xu, W., Li, X., & Zheng, X. (2021). Improved Visual Recognition Memory Model Based on Grid Cells for Face Recognition. Frontiers in Neuroscience, 15, 718541.

---

### Meta-Review · Area_Chair_UrVG · 2026-01-06

**Summary:**

This paper proposes an unsupervised local learning principle for spiking neural networks that combines latency mediated spike timing dependent plasticity with a proximity learning mechanism inspired by volume transmission. Applied to a bio inspired sound source localization model, the method demonstrates very high localization accuracy and fine angular resolution. Reviewers acknowledge the strong biological motivation and the effectiveness of the approach on the target task, while opinions differ on the broader impact and generality of the contribution.

**Reviewer Concerns:**

Reviewers raise concerns about the limited scope and generality of the work, noting that the learning rules are closely tied to the Jeffress model and evaluated only on a highly controlled sound localization task. Several reviewers question the conceptual novelty of combining existing biological mechanisms and request clearer evidence that the approach extends beyond this specific application. Additional concerns include limited robustness evaluation under realistic noise conditions and the need for clearer comparisons with supervised SNN training methods.

**Reviewer Scores:**

Reviewer scores are mixed. One reviewer rated the paper clearly above the acceptance threshold, while several others rated it as reject or marginal reject. The score distribution reflects agreement on technical soundness and biological relevance, alongside disagreement about novelty, scope, and overall impact.

---

### Decision · Program_Chairs · 2026-01-26

Reject